# ROYAL SOCIETY
# OPEN SCIENCE

## Registered report

psychology

competition, priority rule, metascience, information sampling, incentive structures

**Author for correspondence:**
Leonid Tiokhin
e-mail: l.tiokhin@tue.nl

# Competition for novelty reduces information sampling in a research game - a registered report

## Leonid Tiokhin[1,2] and Maxime Derex[3,4]

[1]School of Human Evolution and Social Change, Arizona State University, Tempe, AZ 85287, USA
[2]Human Technology Interaction Group, Eindhoven University of Technology, PO Box 513, 5600 MB Eindhoven, The Netherlands
[3]Human Behaviour and Cultural Evolution Group, Department of Biosciences, University of Exeter, Penryn TR10 9FE, UK
[4]Laboratory for Experimental Anthropology — ETHICS (EA 7446), Catholic University of Lille, 59016 Lille, France

 LT, 0000-0001-7333-0383; MD, 0000-0002-1512-6496

Incentive structures shape scientists' research practices. One incentive in particular, rewarding priority of publication, is hypothesized to harm scientific reliability by promoting rushed, low-quality research. Here, we develop a laboratory experiment to test whether competition affects information sampling and guessing accuracy in a game that mirrors aspects of scientific investigation. In our experiment, individuals gather data in order to guess true states of the world and face a tradeoff between guessing quickly and increasing accuracy by acquiring more information. To test whether competition affects accuracy, we compare a treatment in which individuals are rewarded for each correct guess to a treatment where individuals face the possibility of being 'scooped' by a competitor. In a second set of conditions, we make information acquisition contingent on solving arithmetic problems to test whether competition increases individual effort (i.e. arithmetic-problem solving speed). We find that competition causes individuals to make guesses using less information, thereby reducing their accuracy (*H1a* and *H1b* confirmed). We find no evidence that competition increases individual effort (*H2*, inconclusive evidence). Our experiment provides proof of concept that rewarding priority of publication can incentivize individuals to acquire less information, producing lower-quality research as a consequence.

## 1. Introduction

A central aim of science is generating reliable results to produce increasingly accurate theories about the world. However,

reliability cannot be taken for granted. Models of the scientific process suggest that, given current incentive structures, many research findings will be false [1–4]. Empirical findings from large-scale replication efforts in diverse fields corroborate this conclusion: many results fail to replicate [5–9]. Science may be our most powerful tool for generating knowledge, but there is clearly room for improvement.

Many factors plausibly affect the reliability of science [10]. At the heart of these are incentive structures: by determining the professional payoffs for various types of research, incentives shape scientists' research decisions [11]. Many current incentives are thought to harm the efficiency of science, including publication bias in favour of positive and novel findings, evaluating scientists based on number of publications and lack of rewards for data sharing and transparent research [10]. In recent years, scholars have been especially concerned about the harmful effects of competition on the scientific process [10–18]. Competitive incentive structures are circumstances in which individuals can expend finite resources (e.g. time, money) to increase their probability of receiving a payoff, and one individual's success reduces the probability that others will succeed [19]. By this definition, much of science is competitive: scientists compete for publication in journals, limited professional positions and funding opportunities [11,12,18,20].

Competition over priority of discovery is arguably one of the most important forms of competition in science. Academic science has a longstanding norm of rewarding individuals for making discoveries and publishing novel findings. Over 50 years ago, sociologist of science Robert Merton noted how this norm might benefit science: rewarding priority can incentivize scientists to invest effort to quickly solve important problems and share their discoveries with the scientific community [21]. Models of academic priority races substantiate Merton's intuition: under some conditions, rewarding priority of discovery can incentivize the disclosure of partial results [22–25] and lead to efficient distributions of scientists across research problems [26]. Nonetheless, scholars have also had longstanding concerns about the repercussions of this norm. Charles Darwin thought that rewarding priority by naming species after their first-describers incentivized biologists to produce 'hasty and careless work' by 'miserably describing a species in two or three lines' [21, p. 644]. More recently, concerns over the consequences of rewarding priority have led the academic journals *eLife* and *PLOS Biology* to offer 'scoop protection' (i.e. allowing researchers to publish findings identical to those already published in the same journal) in attempts to reduce the disproportionate payoffs to scientists who publish first [27–29]. In the editorial justifying their new policy, The *PLOS Biology* Staff Editors write '…many of us know researchers who have rushed a study into publication before doing all the necessary controls because they were afraid of being scooped. Of course, healthy competition can be good for science, but the pressure to be first is often deleterious…' [28].

Despite these reasonable concerns, there is little empirical evidence for the hypothesis that competitive pressures to publish cause individuals to produce lower-quality research. In focus-group discussions with mid- and early-career researchers, scientists acknowledge that competition incentivizes them to conduct careless work [12], but laboratory experiments investigating competition more broadly demonstrate that competition also promotes individual effort [19,20,30–33]. As a consequence, it is unclear how competition in general, and competition for priority in particular, affects research quality. On the one hand, competition might cause researchers to make dubious claims based on inadequate data. On the other, competition might encourage researchers to gather data more efficiently.

Given the difficulty of experimentally manipulating incentives in real-world scientific practice, we developed a simple game that mimics aspects of scientific investigation. In our experiment, participants must gather data in order to guess true states of the world and face a tradeoff between guessing quickly and increasing accuracy by acquiring more information. Although this game is a simplification of the scientific process, leaving out many factors that exist in real-world scientific research, it allows us to investigate two hypothesized effects of competition on information-sampling strategies in controlled conditions. By doing so, our experiment brings quantitative data to the debate about whether competition necessarily causes individuals to sacrifice research quality by trading accuracy for speed or whether individuals can avoid this tradeoff by modulating their effort.

## 1.1. Study aims

We develop a simple experiment to test the effect of competition for priority on information-acquisition strategies. To do so, we modify the Cambridge Information Sampling Task (IST) [34] to create a game in which individuals gather information in order to guess true states about the world. Participants are

incentivized to make as many correct guesses as possible and face a tradeoff between guessing quickly and increasing accuracy by gathering more information.

In order to investigate the effect of competition on how individuals acquire information, we compare a baseline treatment in which participants are rewarded for each correct guess to a treatment where participants face the possibility of being 'scooped' by a competitor that makes the correct guess more quickly. In a second set of treatments, we make the rate at which individuals can acquire information contingent on individuals' effort. Doing so allows us to test whether competition to guess first leads to a faster rate of information acquisition.

This set of treatments investigates two potential effects of competition on information acquisition: a negative effect on reliability (individuals might make inferences from smaller amounts of evidence) and a positive effect on productivity (individuals might work to acquire evidence at a faster rate). Such a design allows us to test whether competition necessarily encourages individuals to trade accuracy for speed or whether individuals can avoid this tradeoff by adjusting their level of effort.

Below, we outline the experimental design and hypotheses and present a simple analytical model based on the experimental parameters. We then outline all planned analyses and pilot-study results before presenting results from the main experiment.

# 2. Material and methods

## 2.1. Game design

Our computer game (a modified version of the Cambridge Information Sampling Task [34]; programmed in Object Pascal with Delphi 7) provides a simple instantiation of a process in which individuals decide how much information to gather when solving a problem and face tradeoffs between quickly producing an answer and increasing accuracy via larger samples. Participants view a screen with 25 black tiles arranged in a $5 \times 5$ grid (figure 1a). Each tile has one of two underlying colours (yellow or blue) and participants can click any tile to reveal its underlying colour. Participants must guess the grid's majority (i.e. most common) colour and are rewarded for accurate guesses. After guessing, participants move on to the next grid. Participants are informed whether their guess was correct or incorrect and can see their cumulative score. In the No-Competition treatments (see Treatments), participants play 20 min and so face a speed–accuracy tradeoff: guessing earlier (i.e. with few tiles clicked) allows them to quickly move on to subsequent grids but decreases their probability of guessing correctly.

The proportion of yellow and blue tiles and their order of appearance are deterministic, but remain unknown to participants. This controls for stochasticity in access to information by ensuring that each participant receives the same information in the same order, regardless of the tiles that a participant clicks. Each grid is characterized by a proportion of yellow and blue tiles (i.e. effect size). This proportion is chosen randomly from one of three possible ratios ($8:17$, $10:15$, $12:13$) and yellow and blue tiles have the same probability of being in the majority. To control for stochasticity in the grids solved by different participants, all participants receive the same grids in the same order. Because one of the two colours might be more salient for the human visual system (e.g. 4 blue : 2 yellow might be a stronger visual signal for blue than 4 yellow : 2 blue is a signal for yellow), the baseline colour is randomly selected at the beginning of the experiment for each participant (e.g. some participants see Y–Y–B–Y–...–Y and have to guess Y, while others see B–B–Y–B–...–B and have to guess B). This aims to limit unforeseen bias.

## 2.2. Treatments

We use a $2 \times 2$ between-subjects design to investigate two treatments (No-Competition; Competition) and two conditions (No-Effort; Effort).

## 2.3. No-Effort condition

### 2.3.1. No-Competition

Participants play the game for 20 min and can acquire information by clicking one tile every 1 s. The 1 s delay between clicking tiles prevents participants from increasing their clicking speed to acquire

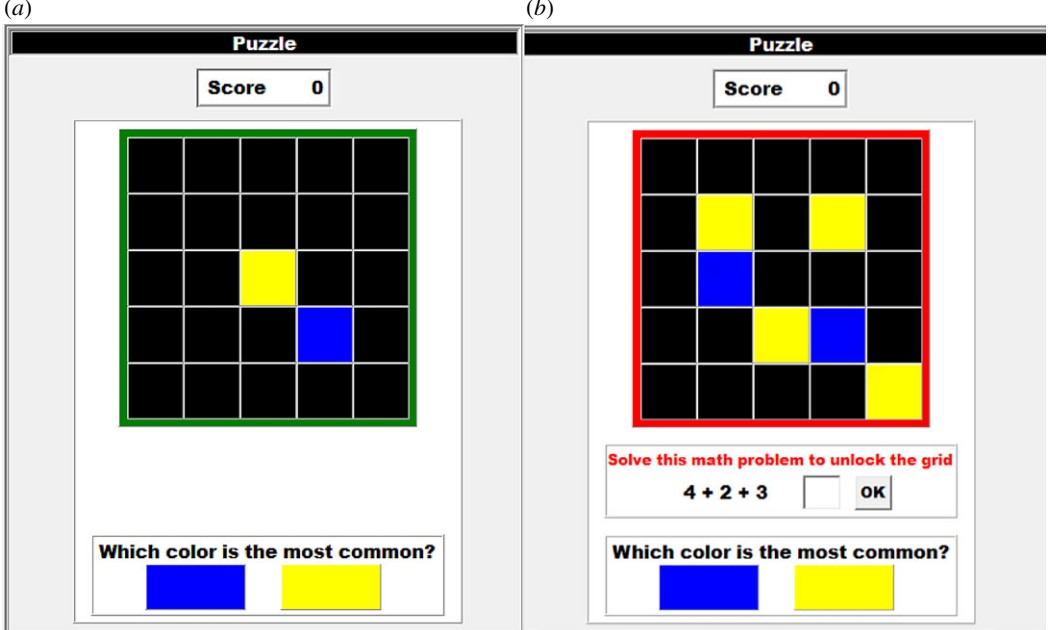

**Figure 1.** Game principle. The experimental task consists of 25 black tiles arranged in a 5 × 5 grid. Participants can click on tiles to reveal their underlying colour (yellow or blue) and are rewarded for correct solutions (i.e. correctly guessing the majority colour). Clicking more tiles provides more information about the grid but takes more time. The order in which colours are revealed is deterministic (i.e. independent of which tile is clicked) but remains unknown to participants. (a) In the No-Effort condition, participants can click a black tile every 1 s. (b) In the Effort condition, participants need to solve an arithmetic problem to acquire information (i.e. click a tile). In the No-Competition treatment, participants gain/lose one point for each correct/incorrect guess. In the Competition treatment, participants only gain/lose points if they guess sooner than their competitor.

information faster than one tile per second. Participants receive payoffs solely as a function of their own performance: they gain one point for each correct guess and lose one point for each incorrect guess. This payoff structure incentivizes participants to acquire at least some information before guessing, because guessing without clicking any tiles results in a 50% probability of answering correctly and hence an expected payoff of 0.

### 2.3.2. Competition

Participants compete against the performance of one previous same-sex participant from the No-Competition treatment. These competitors will be sampled without replacement: each participant will compete against the performance of one unique previous participant. After submitting their guess, participants move on to the subsequent problem, where they again compete against the same participant's performance on that problem until they solve the same set of grids as their competitor. For grids where the competitor guesses correctly, participants receive a payoff only if they are correct and guess faster than their competitor. For grids where the competitor guesses incorrectly, participants receive a payoff solely as a function of their own performance. Participants are informed of their payoff at the end of each round and are notified whenever they guess after a competitor who has already guessed correctly.

We instantiate competition by having participants compete against the performance of baseline participants for several reasons. A more realistic instantiation would be for two participants in the Competition treatment to directly compete against each other. In this design, for both participants to remain synchronized, they would need to move to the next grid as soon as either participant guessed. Such a design is problematic: it only generates data from one participant per grid (i.e. the guesser) and that data are biased towards the participant that reacts most to competition by guessing earliest. One potential solution is to synchronize participants by allowing both participants to make their guess on each grid. However, this introduces other problems: the participant that guesses earlier is forced to wait for their competitor to guess. As a consequence, the two competitors would not experience the exact same experimental conditions, which may result in experimental biases. For example, participants might delay their guesses to avoid waiting for their competitor to guess. Our design avoids these problems.

## 2.4. Effort condition

Treatments in the Effort condition are identical to treatments in the No-Effort condition, except that participants need to solve a simple arithmetic problem before being able to click on a tile (figure 1b). This is a commonly used real-effort task in economics [35]. In this treatment, participants can increase their rate of clicking tiles by solving arithmetic problems more quickly (See Pilot study for checks on floor effects).

## 2.5. Sampling plan

Arizona State University students (equal numbers of women and men, age 18 and over) will be randomly selected from a database managed by the Elinor Ostrom Multi-Method Lab at Arizona State University and recruited by email. We will obtain informed consent from all subjects before the experiment's onset (ethical approval has been obtained from the Arizona State University Institutional Review Board, code: STUDY00007691). Participants will receive $5 for participation and an additional amount ranging from $0 to $10 depending on their performance (see Score calculation).

## 2.6. Procedure

The experiment will take place in a computer room at Arizona State University. Each session will consist of a maximum of 16 participants (exclusively male or female). Within sessions, each participant will be assigned to either the Competition or No-Competition treatment. Participants in the Competition treatment will be randomly assigned to compete against the performance of one same-sex participant who had previously participated in an equivalent No-Competition treatment. This requires that all participants in the first experimental session be assigned to the No-Competition treatment. Within subsequent sessions, participants will be assigned to the Competition treatment if there exists a previous participant against whom they can compete and will be assigned to the No-Competition treatment otherwise.[1] Within each session, each participant will be randomly assigned to either the Effort or No-Effort condition. All manipulations (Competition versus No-Competition; Effort versus No-Effort) are between subjects.

Participants will enter the computer room in the order that they arrive to the experiment, will sit at physically separated computers and will be instructed that communication is not allowed. Participants will be blind to the fact that there are four experimental treatments. Before starting the experiment, participants will be requested to enter their age and sex. Participants will be shown the instructions on their screens. In the No-Competition treatments, the game will last 20 min. In the Competition treatments, the game will last as long as it takes participants to complete the same set of grids that was completed by their competitor. At the end of the game, participants will receive a reward according to their performance (see Score calculation).

## 2.7. Tutorial and pre-game information

Before starting, participants will complete a tutorial in which they will perform basic actions. This tutorial will guide participants' actions so that participants experience clicking tiles and choosing the majority colour, to ensure that all participants have mastered the interface before playing. Participants in the No-Competition treatments will be informed that the goal of the game is to make as many correct guesses as possible within the duration of the experiment. Participants in the Competition treatments will be informed that they are playing against the performance of another participant and that the goal of the game is to be the first to make as many correct guesses as possible. Participants will be informed that their score and monetary reward will depend on their total number of correct guesses and correct faster-guesses, respectively. In the Effort condition, participants will experience solving an arithmetic problem. Participants will not be informed about the total number of yellow and blue tiles (i.e. effect size) per round.

---

[1]Before data collection, we requested and received editorial approval for a minor deviation from the protocol approved in Stage 1: instead of sessions where all participants are assigned to either the No-Competition or Competition treatment, we ran sessions such that participants could be assigned to either treatment within each session.

## 2.8. Score calculation

In the No-Competition treatments, participants will receive 1 point for each correct guess and will lose 1 point for each incorrect one. In the Competition treatments, the payoff structure is identical when participants guess (i) sooner than their competitor, or (ii) after a competitor who guessed incorrectly. Participants do not gain or lose any points when guessing after a competitor who guessed correctly. This payoff structure corresponds to the assumption that being scooped prevents researchers from both receiving a benefit for being correct and paying a cost for being wrong. In the unlikely event that a participant guesses at the exact same time as their competitor, they will receive 1 point. A participant's final score will be the sum of these points. The function that translates scores into payoffs will remain unknown to participants,

$$\text{Payoff}_{\text{no-competition}} = \$0.15 \times \text{CorrectGuesses} - \$0.15 \times \text{IncorrectGuesses}$$
$$\text{Payoff}_{\text{competition}} = \$0.15 \times \text{CorrectFasterGuesses} - \$0.15 \times \text{IncorrectFasterGuesses}.$$

## 2.9. Data-collection stopping rules

We specify a region of practical equivalence (ROPE) for all relevant parameters ([36], see 'Analyses and predictions'). We will stop data collection after the 95% highest probability density interval (HPDI) for all parameters falls entirely inside or outside pre-specified ROPEs for each hypothesis. We will check whether the HPDIs fall inside or outside each ROPE after every four sessions of data collection (one session corresponding to each of the four treatments; maximum 16 participants per session) and will collect data until we obtain a maximum of 260 participants (an upper limit set by funding availability). Data from participants excluded based on pre-specified criteria will not count towards this 260-participant limit.

## 2.10. Completion timeline

If Stage 1 review is successful, we anticipate completing the experiment and submitting the manuscript for Stage 2 review within five months of receiving stage 1 approval.

# 3. Model

We developed a simple mathematical model to better understand the payoff structure of our experiment and to gain insight into how competition should affect participants' behaviour in the No-Effort condition. The goal of this model is to understand whether *H1a* (see Hypotheses) is logically coherent (i.e. that our instantiation of competition for priority actually incentivizes participants to guess with smaller amounts of evidence).

In the experiment's Competition treatments, participants who guess the underlying colour only gain or lose one point if they (i) take less time to guess than their opponent or (ii) take longer to guess than their opponent, but their opponent has guessed incorrectly. When a participant guesses before or at the same time as an opponent, the participant's expected payoff (EP) is

$$\text{EP} = p_{\text{P}} - (1 - p_{\text{P}})$$
$$\text{EP} = 2p_{\text{P}} - 1$$

where $p_{\text{P}}$ is the probability that the participant guesses correctly. When a participant guesses after their opponent, the participant's EP is

$$\text{EP} = 0p_{\text{o}} + p_{\text{P}}(1 - p_{\text{o}}) - (1 - p_{\text{P}})(1 - p_{\text{o}})$$
$$\text{EP} = 2p_{\text{P}} + p_{\text{o}} - 2p_{\text{P}}p_{\text{o}} - 1$$

where $p_{\text{o}}$ is the probability that the participant's opponent (i.e. a participant in the No-Competition treatment) guesses correctly. This assumes that guessing at the same time or before an opponent is equivalent, and that only guessing after an opponent results in some probability of being scooped. EP can take on values between 0 and 1, because participants have a minimum 0.5 probability of correctly guessing the underlying colour, and payoffs to correct and incorrect guesses are symmetrical. $p_{\text{P}}$ and $p_{\text{o}}$ are a function of two parameters: the ratio of coloured tiles (i.e. effect size $8:17$, $10:15$ or $12:13$) and the number of tiles that a participant/opponent reveals (we assume that participants' time to guess is entirely determined by the number of tiles they reveal). To calculate $p_{\text{P}}$ and $p_{\text{o}}$, we simulated the average amount of information available to a participant, conditional on the effect size and the participant

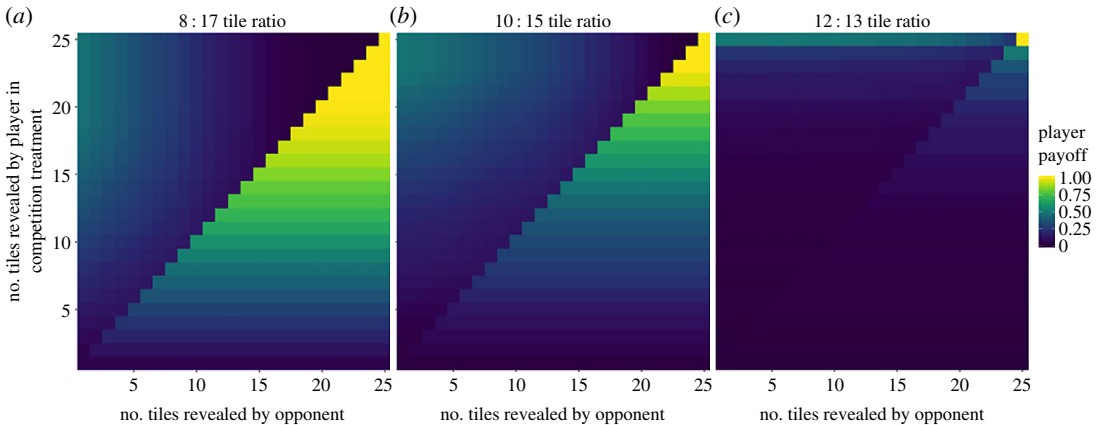

**Figure 2.** A participant's expected payoff as a function of the number of tiles revealed by the participant and their competitor. Plotted ($a-c$) for three ratios of coloured tiles (i.e. effect sizes): 8 : 17, 10 : 15 and 12 : 13. $X$- and $Y$-axes indicate the number of tiles revealed by a participant's opponent and the participant, respectively. A participant's expected payoff is highest when their opponent reveals a large number of tiles and the participant reveals the exact same or fewer tiles. The largest drop in payoff occurs when a participant reveals slightly more tiles than their opponent. When an opponent guesses after revealing few tiles, a participant maximizes their expected payoff by revealing many tiles before guessing.

revealing a given number of tiles (electronic supplementary material, figure S1; code available here: https://osf.io/udm8g/). For each effect size, we randomly generated 25-tile sequences of yellow and blue tiles (500 simulations). We then computed the proportion of all possible 25-tile sequences that give the same outcome as the current majority colour after $n$ tiles have been revealed. For example, consider a participant who reveals a 7-tile sequence of Y–Y–Y–Y–B–Y–Y. Given this initial sequence of tiles, there are 230 964 sequences of 25 tiles, out of all possible sequences, that result in a majority yellow colour and 31 180 sequences that result in a majority blue colour. The amount of information available to this participant is $230\,964/(31\,180 + 230\,964) = 0.881$. This means that if the participant guesses that yellow is the majority colour from that specific 7-tile sequence, the participant will be correct 88.1% of the time.

Figure 2 plots EP as a function of the number of tiles that a participant and their opponent reveal, for different effect sizes. If an opponent reveals many tiles, a participant receives the highest EP by revealing the exact same number (or slightly fewer) numbers of tiles. This occurs because participants have a high probability of guessing the majority colour when they reveal a large number of tiles and guessing before an opponent guarantees that the participant does not get scooped. If a participant guesses after an opponent who has revealed many tiles, a participant's EP is low: the opponent will usually correctly guess the majority colour, causing the participant to obtain 0 points. If an opponent reveals very few tiles, a participant receives the highest EP by revealing a large number of tiles. This occurs because a participant who guesses before this opponent has a high probability of guessing incorrectly, whereas a participant who guesses after this opponent can maximize their probability of correctly guessing the majority colour by revealing as many tiles as possible. As effect sizes decrease, a participant receives a higher EP by revealing more tiles. This occurs because, for most possible numbers of tiles revealed below 25, a participant has a lower probability of guessing correctly when effect sizes are small.

To assess the logical coherence of *H1a* (see Hypotheses), we calculated participants' EP as a function of the effect size and number of tiles revealed, assuming that participants compete against payoff-maximizing competitors (i.e. individuals who reveal the number of tiles that maximizes their EP in the No-Competition × No-Effort treatment). The results corroborate the intuition suggested by a visual inspection of figure 2: when competing against payoff-maximizing competitors (who reveal large numbers of tiles), participants maximize EP by revealing the same number or fewer tiles than their competitor (electronic supplementary material). Sensitivity checks indicate that this result is robust to incorporating stochasticity in the number of tiles revealed by competitors.[2]

---

[2]After receiving Stage 1 approval, we realized a peculiarity of this model. The model determined participants' expected payoff, conditional on participants assuming that small differences in the ratio of yellow to blue tiles are much more probable than large ones. This is equivalent to assuming that participants enter the experiment with a strong prior belief that effects are small. In the electronic supplementary material, we modify this assumption by determining the expected number of tiles revealed by simulated Bayesian participants who begin with a flat prior across possible effect sizes. This alternative model produces qualitatively similar results (electronic supplementary material, figures S4–S6): participants in the No-Competition treatment maximize their expected

This model provides support for the logical coherence of *H1a*: the experiment's payoff structure incentivizes participants to guess at the same time as or before their competitor. However, the former outcome is unlikely: in the experiment (unlike the model), priority is determined by the amount of time spent before guessing, not by the number of tiles revealed. As such, participants in the Competition × No-Effort treatment usually maximize their EP by guessing their competitor.

# 4. Hypotheses

## 4.1. Condition 1: No-Effort

*H1a: Competition for priority will cause participants to guess with smaller amounts of evidence.*

In the first set of experimental treatments, individuals cannot acquire information faster than a predetermined rate (figure 1*a*). As a consequence, when individuals are in competition to guess correctly before a competitor, they can only do so by making their guess with less information (i.e. revealing less tiles before guessing).

*H1b: If H1a is confirmed, then competition for priority will also cause participants in Condition 1 to have reduced accuracy.*

If participants reveal less tiles before guessing, then they will have a lower probability of making a correct guess on each grid, because the probability of making a correct guess is a monotonically increasing function of the number of tiles revealed.

## 4.2. Condition 2: Effort

*H2: Competition for priority will increase participant' effort, thereby causing participants to reveal information faster.*

In the second set of experimental treatments, the rate at which information can be acquired depends on individuals' effort: individuals need to solve an arithmetic problem for each piece of information (figure 1*b*). Individuals can thus affect their rate of information acquisition by adjusting their effort (i.e. solving arithmetic problems faster or slower). Research in experimental economics has found that competition increases participants' effort in similar tasks [19,35]. As such, we expect competition for priority to cause participants to solve arithmetic problems at a faster rate (*H2*).

## 4.3. Interaction between Condition 1 and Condition 2

*H3a: The effect of competition for priority on reducing the amount of evidence that participants gather will be larger in Condition 1 than in Condition 2.*
*H3b: If H3a is confirmed, then competition will cause a bigger reduction in participant accuracy in Condition 1 than in Condition 2.*

In the Effort condition, individuals can potentially guess before their competitor in two ways: reveal less information (i.e. fewer tiles) before guessing or increase effort and solve arithmetic problems more quickly. These are not mutually exclusive. However, if individuals do increase their speed of solving arithmetic problems, then we expect competition to have a smaller effect on their accuracy. We make no prediction about whether competition will have absolutely no effect on participant accuracy in the Effort condition or whether competition will simply have a smaller, negative effect on accuracy, compared to the No-Effort condition.

# 5. Analyses and predictions

We will fit statistical models within a Bayesian framework with weakly informative priors, using map2stan in the *rethinking* package in R [37,38]. Table 1 ('Priors') lists prior probability distributions for parameters in all statistical models.

payoff by guessing sooner when effect sizes are larger. Further, when playing against a payoff-maximizing competitor, participants in the Competition treatment maximize their expected payoff by guessing at the exact same time or before their competitor.

## 5.1. Power analysis

To check whether 260 participants provide sufficient statistical power to evaluate our hypotheses, we used our pilot data (see Pilot Study), to conduct a power analysis. For confirmatory analyses, we set a ROPE for each relevant parameter by determining the minimum effect size that could be detected 95% of the time, given our maximum sample size of 260 participants. This minimum effect size then determined the upper and lower bounds of the ROPE for each analysis [39,40]. We considered an alternative approach: setting all ROPE boundaries based on theoretical considerations [36]. However, because our hypotheses only make directional predictions, they provide no guidance as to an effect's size or minimum effect sizes of interest.

Our pilot data allowed us to conduct power analyses for *H1a,b* and *H3a,b*. For *H2* (Model 3), we did not record participants' time between clicking one tile and being allowed to click a subsequent tile (i.e. time to accurately solve one arithmetic problem). As such, we instead conducted a power analysis for Model 3 using a slightly different outcome variable: time to produce any answer (accurate or inaccurate) for one arithmetic problem. We then set a ROPE based on 99% statistical power, to compensate for the uncertainty introduced by basing the power analysis on a statistical model with a different outcome variable.

For each power analysis, we followed the following steps (code available at https://osf.io/udm8g/):

1. Analyse the pilot data with Bayesian statistical models using map2stan in the *rethinking* package in R (see Analyses and Predictions) [37,38].
2. Extract samples for all parameters from the posterior probability distribution for a given statistical model.
3. Simulate 260 participants with $x$ observations per participant, where $x$ is randomly sampled (with replacement) from the number of observations per participant in the pilot.
4. Generate simulated data for each participant by taking random samples from the posterior probability distribution for each parameter and inserting those samples into the formula implied by the statistical model structure for an analysis.
5. Analyse the simulated data with a frequentist implementation of the statistical model in the proposed analysis.
6. Record the confidence interval for each parameter of interest.
7. Repeat steps 2–6, 500 times.
8. Generate a ROPE for each parameter of interest by determining the maximum effect size that fell outside of the 95% confidence interval in the 500 simulations.

## 5.2. Testing hypotheses

For quality checks, we generated a ROPE for each parameter based on our subjective assessment of what effect size would convincingly indicate that a manipulation was successful. For confirmatory predictions, we generated a ROPE for each parameter by conducting power analyses to determine the minimum effect size that can be detected 95% of the time, given our maximum sample size.

If the 95% HPDI for a parameter falls outside of the ROPE, we will consider this as evidence against the null hypothesis of no effect. If the 95% HPDI for a parameter falls outside of the ROPE and is in the direction predicted by a hypothesis, we will consider this as evidence for the hypothesis. If the 95% HPDI for a parameter falls outside of the ROPE and is in the opposite direction to that predicted by a hypothesis, we will consider this as evidence against the hypothesis. If the 95% HPDI falls within the ROPE, we will consider this as evidence for the null hypothesis of no effect. If the 95% HPDI does not fall entirely within or outside the ROPE, we will consider that the study does not provide conclusive evidence for either the prediction or the null hypothesis.

## 5.3. Exclusions and outliers

We will exclude all data from participants who did not complete the study (i.e. who did not answer the final 'Competition attention check' question; see Quality checks). We will also exclude all data from participants who inform the experimenter of technical difficulties during the study.[3] Within individual

---

[3]While data collection was ongoing, we requested and received editorial approval for this minor addition to the exclusion criteria approved in Stage 1. We did not analyse the data using this additional exclusion criterion until receiving editorial approval.

**Table 1.** Prior probability distributions for all statistical models, including quality checks (Effort manipulation check, Competition attention check) and confirmatory analysis plans (Models 1–3). Gamma distributions are defined by parameters for shape and rate. Normal distributions are defined by parameters for mean and standard deviation.

| parameter | Effort manipulation check | Competition attention check | Model 1 (tiles) | Model 2 (correct guess) | Model 3 (arithmetic time) |
|---|---|---|---|---|---|
| $\sigma$ | gamma (2, 0.5) | n.a. | gamma (2, 0.5) | n.a. | gamma (2, 0.5) |
| $\alpha$ | gamma (1.5, 0.05) | normal (0, 10) | uniform (0, 25) | normal (0, 10) | gamma(1, 0.05) |
| $\alpha_{PARTICIPANT}$ | normal (0, $\sigma_{PARTICIPANT}$) | n.a. | normal (0, $\sigma_{PARTICIPANT}$) | normal (0, $\sigma_{PARTICIPANT}$) | normal (0, $\sigma_{PARTICIPANT}$) |
| $\sigma_{PARTICIPANT}$ | gamma (1.5, 0.05) | n.a. | gamma (1.5, 0.05) | gamma (1.5, 0.05) | gamma (1, 0.05) |
| $\beta_C$ | normal (0, 10) | normal (0, 10) | normal (0, 10) | normal (0, 10) | normal (0, 10) |
| $\beta_E$ | normal (0, 10) | n.a. | normal (0, 10) | normal (0, 10) | n.a. |
| $\beta_{CE}$ | normal (0, 10) | n.a. | normal (0, 10) | normal (0, 10) | n.a. |
| $\beta_{Ns}$ | n.a. | n.a. | normal (0, 10) | normal (0, 10) | normal (0, 10) |

participants, we will exclude observations for which there is missing data for at least one measured variable. Both participants' time to make a guess and time to solve arithmetic problems follow heavily right-skewed distributions (see electronic supplementary material). For participants' time to make a guess, we will exclude times that are more than 5 s.d. larger than the mean time until making a guess. For participants' time to solve arithmetic problems, we will exclude arithmetic-problem solving times that are more than 5 s.d. larger than the mean arithmetic-problem solving time. These criteria allow for the exclusion of the most extreme data points while preventing the exclusion of too many observations. These same exclusion criteria are also used in the analysis of the pilot data (electronic supplementary material).

## 5.4. Quality checks

### 5.4.1. Effort manipulation

If the Effort manipulation is successful, then participants in the Effort condition should take longer per click than participants in the No-Effort condition. To assess the effect of effort on the average time to click a tile, we will use a linear regression, with random effects for the participant, of the following form:

$$Y_i \sim \text{Normal}(\mu_i, \sigma)$$
$$\mu_i = \alpha + \alpha_{\text{PARTICIPANT}i} + \beta_E E_i + \beta_C C_i + \beta_{CE} C_i E_i,$$

$Y_i$: time (s) to click one tile and reveal its underlying colour. $\alpha$: intercept. $\alpha_{\text{PARTICIPANT}i}$: random intercept for each participant. $E$: Effort condition (1/0). $C$: Competition treatment (1/0). $CE$: Interaction between treatment and effort.

### 5.4.2. Competition attention check

At the end of the experiment, participants will be asked 'During the experiment, were you competing with another player to be first to guess the correct answer?'. If the participants in the Competition treatment are aware that they competed against another individual, then a higher proportion of participants in the Competition treatments should answer 'yes' to this question than in the No-Competition treatments. To assess the effect of competition on answering 'yes' to this question, we will use a logistic regression of the following form:

$$Y_i \sim \text{Binomial}(1, p_i)$$
$$\text{Logit}(p_i) = \alpha + \beta_C C_i$$

$Y_i$: answered 'yes'. $\alpha$: intercept. $C$: Competition treatment (1/0).

## 5.5. Confirmatory analysis plans

*H1a: Competition for priority will cause participants in the No-Effort condition to guess with smaller amounts of evidence.*
*H3a: Competition for priority will cause a bigger reduction in the amount of evidence that participants gather in the No-Effort condition than in the Effort condition.*

We will use one dependent measure to test whether competition causes participants to guess with smaller amounts of evidence: the number of tiles that participants reveal when making a guess.

In the Competition treatments, participants should reveal fewer tiles when making their guess than in the No-Competition treatments. There should also be a positive interaction between the Competition treatments and the Effort condition: the effect of competition on the number of tiles revealed should be smaller in the Effort condition than in the No-Effort condition (due to the expected positive effect of competition on individual effort).

*Model 1*: To compare the number of tiles that participants reveal before guessing the majority colour in the Competition treatments versus the No-Competition treatments, we will use a multiple linear regression, with random effects for each participant, of the following form:

$$Y_i \sim \text{Normal}(\mu_i, \sigma)$$
$$\mu_i = \alpha + \alpha_{\text{PARTICIPANT}i} + \beta_C C_i + \beta_E E_i + \beta_{CE} C_i E_i + \beta_{Ns} Ns_i$$

$Y_i$: number of tiles clicked before guessing. $\alpha$: intercept. $\alpha_{\text{PARTICIPANT}i}$: random intercept for each

**Table 2.** Region of practical equivalence (ROPE) for quality checks and confirmatory analyses. ROPEs for quality checks are based on subjective assessment of what effect size would convincingly indicate a successful manipulation. ROPEs for confirmatory analyses (Models 1–3) are based on 95% statistical power unless indicated otherwise. Model 1 tests the effect of the Competition treatment and Effort condition on number of tiles clicked before guessing, using a multiple linear regression with random effects for each participant. Model 2 tests the effect of the Competition treatment and Effort condition on the probability of a correct guess, using a logistic regression with random effects for each participant. Model 3 tests the effect of the Competition treatment on the time to accurately solve one arithmetic problem, using a multiple linear regression with random effects for each participant.

| parameter | Effort manipulation check | Competition attention check | Model 1 (tiles) | Model 2 (correct guess) | Model 3 (arithmetic time) |
|---|---|---|---|---|---|
| $\beta_C$ | n.a. | (−0.8, 0.8) | (−1.22, 1.22) | (−0.19, 0.19) | (−0.33, 0.33)[b] |
| $\beta_E$ | (−0.5, 0.5) | n.a. | n.a. | n.a. | n.a. |
| $\beta_{CE}$ | n.a. | n.a. | (−0.10, 0.10)[a] | (−0.09, 0.09) | n.a. |

[a]ROPE based on 85% statistical power.
[b]ROPE based on 99% statistical power.

participant. $C$: Competition treatment (1/0). $E$: Effort condition (1/0). $CE$: interaction between treatment and effort. $Ns$: standardized number of tiles for the majority colour (i.e. effect size).

*H1b: If H1a is confirmed, then competition for priority will also cause participants in the No-Effort condition to have reduced accuracy.*
*H3b: If H3a is confirmed, then competition for priority will cause a bigger reduction in participant accuracy in the No-Effort condition than in the Effort condition.*

We will use one dependent measure to test the effect of competition for priority on accuracy: the probability of correctly guessing the majority colour. In the Competition treatments, participants should have a lower probability of making correct guesses than in the No-Competition treatments. There should also be a positive interaction between the Competition treatments and the Effort condition: the effect of competition on accuracy should be smaller in the Effort condition than in the No-Effort condition.

*Model 2:* To assess the probability of a correct guess, we will use a logistic regression, with random effects for the participant, of the following form:

$$S_i \sim \text{Binomial}(1, p_i)$$
$$\text{Logit}(p_i) = \alpha + \alpha_{\text{PARTICIPANT}_i} + \beta_C C_i + \beta_E E_i + \beta_{CE} C_i E_i + \beta_{Ns} Ns_i$$

$S_i$: successful guess. $\alpha$: intercept. $\alpha_{\text{PARTICIPANT}i}$: random intercept for each participant. $C$: Competition treatment (1/0). $E$: Effort condition (1/0). $CE$: interaction between treatment and effort. $Ns$: standardized number of tiles for the majority colour (i.e. effect size).

*H2: Competition for priority will increase participant effort, thereby causing participants to reveal information faster.*

We will use one dependent measure to test the effect of competition for priority on effort: amount of time (seconds) between when participants reveal a piece of information (i.e. click a tile) and when participants are able to click the next tile (i.e. the time to accurately solve an arithmetic problem). This analysis will be limited to participants in the Effort condition.

Participants in the Competition × Effort treatment should solve arithmetic problems faster than participants in the No-Competition × Effort treatment. The time that it takes participants to solve arithmetic problems is the time between clicking one tile and being allowed to click the subsequent tile. As a result, participants in the Competition × Effort treatment should have a smaller time between clicking one tile and being allowed to click the subsequent tile compared to participants in the No-Competition × Effort treatment.

*Model 3:* To test the effect of competition on the time between clicking one tile and being allowed to click the subsequent tile (i.e. time to accurately solve one arithmetic problem), we will use a multiple linear regression, with random effects for each participant, of the following form:

$$Y_i \sim \text{Normal}(\mu_i, \sigma)$$
$$\mu_i = \alpha + \alpha_{\text{PARTICIPANT}i} + \beta_C C_i + \beta_{Ns} Ns_i$$

$Y_i$: time between clicking one tile and being allowed to click the subsequent tile (i.e. time to solve an arithmetic problem). $\alpha$: intercept. $\alpha_{\text{PARTICIPANT}i}$: random intercept for each participant. $C$: Competition treatment (1/0). $N$s: standardized number of tiles for the majority colour (i.e. effect size) (table 2).

# 6. Pilot study

We conducted a pre-registered (https://osf.io/udm8g/) pilot study (see electronic supplementary material for detailed results and full exclusion criteria). This study was designed to test the feasibility of the proposed design, not to test hypotheses. In conducting the pilot study, we underspecified exclusion criteria and deviated from the pre-specified pilot analysis plan. We consider all results from the pilot study to be exploratory.

The pilot study involved 48 participants. We excluded data from one participant that did not complete the study. This resulted in a final sample of 47 participants (23 female, 24 male); 16 and 31 participants were assigned to the Competition and No-Competition treatments and 23 and 24 participants were assigned to the Effort and No-Effort conditions, respectively. The pilot design differed from the proposed design in one way: participants were paid $0.25 cents per solution instead of $0.15 cents.

Participants in the Effort conditions spent more time (seconds) per grid (i.e. took longer to guess the majority colour) than participants in the No-Effort conditions (95% HPDI: (11.81, 27.95), mean: 19.66). This provides evidence that the Effort manipulation was successful. Participants in the Competition treatments had a larger log-odds of answering 'yes' to an attention-check question about whether or not they competed against another participant in the experiment (95% HPDI: (3.93, 22.67), mean: 11.92). This provides evidence that participants in the Competition treatments were aware that they were competing against another participant.

Compared to participants in the No-Competition treatments, participants in the Competition treatments revealed fewer tiles (95% HPDI: (−7.46, −0.55), mean: −3.98), did not have a lower log-odds of correctly guessing the majority colour (95% HPDI: (−1.20, 0.03), mean: −0.60), did not spend less time (s) per grid (95% HPDI: (−13.21, 3.92), mean: −5.02) and made a larger number of guesses per minute (95% HPDI: (0.91, 4.12), mean: 2.50). Compared to participants in the No-Competition × Effort treatment, participants in the Competition × Effort treatment solved more arithmetic problems per minute (95% HPDI: (0.55, 4.53), mean: 2.58).

Compared to participants in the No-Effort conditions, participants in the Effort conditions did not reveal fewer tiles (95% HPDI: (−3.35, 2.12), mean: −0.64), did not have a lower log-odds of correctly guessing the majority colour (95% HPDI: (−0.62, 0.45), mean: −0.07), spent more time (s) per grid (95% HPDI: (11.81, 27.95), mean: 19.66) and made a smaller number of guesses per minute (95% HPDI: (−3.36, −0.63), mean: −2.03).

There was no evidence for an interaction between Competition and Effort on the number of tiles revealed (95% HPDI: (−1.51, 7.93), mean: 3.26), log-odds of correctly guessing the majority colour (95% HPDI: (−0.29, 1.53), mean: 0.62) or time (s) spent per grid (95% HPDI: (−13.95, 8.67), mean: −2.32). There was evidence for an interaction between Competition and Effort on the number of guesses made per minute (95% HPDI: (−4.49, −0.13), mean: −2.28): in the No-Effort condition, participants in the Competition treatment made a larger number of guesses per minute than participants in the No-Competition treatment.

# 7. Results

We conducted the experiment according to the in-principle accepted Stage 1 protocol (https://osf.io/24v9k/). Every four sessions of data collection, we checked whether the HPDIs for all parameters fell entirely within or outside the pre-specified ROPEs for each hypothesis. This never occurred. As such, we collected data until we reached the pre-specified maximum sample size of 260 useable participants. In total, we collected data from 269 participants. After applying the pre-specified exclusion criteria, our final sample size was 260 participants (six participants did not complete the study, two participants experienced technical difficulties and one additional participant was excluded because every data point for their time-to-make-a-guess was larger than 5 s.d. from the mean). The final sample was composed of 130 females and 130 males (No-Effort condition: 65 females, 65 males; Effort condition: 65 females, 65 males).

Within individual participants, we excluded observations for which there was no data for at least one measured variable (i.e. when the game ended before participants made their guess). We also excluded observations for time-to-make-a-guess and time-to-solve-arithmetic-problems that were larger than 5

s.d. from their respective means. Below, we present analyses for quality checks and confirmatory predictions, using this final sample of 260 participants. All confirmatory analyses and quality checks use the statistical models specified earlier in this report and approved in Stage 1 (see electronic supplementary materials for Markov chain convergence diagnostics).

## 7.1. Quality checks

Participants in the Effort condition spent more time (s) per click than participants in the No-Effort condition (95% HPDI: (3.26, 3.98), $\beta = 3.61$). The effect falls entirely outside of the pre-specified ROPE of $(-0.5, 0.5)$ and is in the predicted direction. This indicates that Effort manipulation was successful. Participants in the Competition treatments had greater log-odds of answering 'yes' to the Competition attention check question than participants in the No-Competition treatments (95% HPDI: (2.91, 4.29), $\beta = 3.57$). The effect falls entirely outside of the pre-specified ROPE of $(-0.8, 0.8)$ and is in the predicted direction. This indicates that participants in the Competition treatments were aware that they were competing against another participant.

## 7.2. Confirmatory analyses

### 7.2.1. No-Effort condition

*H1a* stated that competition for priority will cause participants in the No-Effort condition to guess with smaller amounts of evidence. Our results provide confirmatory evidence for this hypothesis. Participants in the Competition treatment revealed fewer tiles per grid than participants in the No-Competition treatment (95% HPDI: $(-5.03, -2.39)$, $\beta = -3.70$, figure 3). The effect falls entirely outside of the pre-specified ROPE of $(-1.22, 1.22)$ and is in the predicted direction.

As participants revealed fewer tiles when in competition (*H1a*), we tested whether they exhibited reduced accuracy (*H1b*). Our results provide confirmatory evidence for this hypothesis. Participants in the Competition treatment had a smaller probability of making a correct guess than participants in the No-Competition treatment (95% HPDI: $(-0.11, -0.03)$, $\beta = -0.07$, figure 4). In log-odds, this effect is (95% HPDI: $(-0.64, -0.20)$, $\beta = -0.42$), which falls entirely outside of the pre-specified ROPE of $(-0.19, 0.19)$ and is in the predicted direction.

### 7.2.2. Effort condition

Participants in the Competition treatment revealed fewer tiles than participants in the No-Competition treatment (95% HPDI: $(-4.47, -1.86)$, $\beta = -3.11$, figure 3) and had a smaller probability of making a correct guess (95% HPDI: $(-0.11, -0.02)$, $\beta = -0.07$, figure 4).

*H3a* stated that the effect of competition for priority on tiles revealed and accuracy will be larger in the No-Effort condition than in the Effort condition. Our results provide no conclusive evidence for *H3a* or for the null hypothesis. There was no evidence for a Competition × Effort interaction on the number of tiles revealed (95% HPDI: $(-1.23, 2.54)$, $\beta = 0.60$). The effect does not fall entirely within or outside of the pre-specified ROPE of $(-0.10, 0.10)$. As there was no evidence for the above interaction, we did not expect *H3b* to be confirmed. Indeed, there was no evidence for a Competition × Effort interaction on the log-odds of making a correct guess (95% HPDI: $(-0.30, 0.39)$, $\beta = 0.02$). This does not fall entirely within or outside the pre-specified ROPE of $(-0.09, 0.09)$ and does not provide conclusive evidence for *H3b* or for the null hypothesis.

*H2* stated that competition for priority will increase participant effort, causing participants to solve arithmetic problems faster. Our results do not provide conclusive evidence for this hypothesis or for the null hypothesis. Participants in the Competition × Effort treatment were not faster to accurately solve one arithmetic problem than participants in the No-Competition × Effort treatment (95% HPDI: $(-0.40, 0.35)$, $\beta = -0.02$, figure 5). The effect does not fall entirely within or outside of the pre-specified ROPE of $(-0.33, 0.33)$.

## 7.3. Exploratory analyses

We explored the sensitivity of the results to different outlier-exclusion criteria and alternative statistical models. The qualitative results were robust to these sensitivity checks (electronic supplementary material, tables S1, S2 and figures S7–S11). Our confirmatory analyses found no evidence for several hypotheses

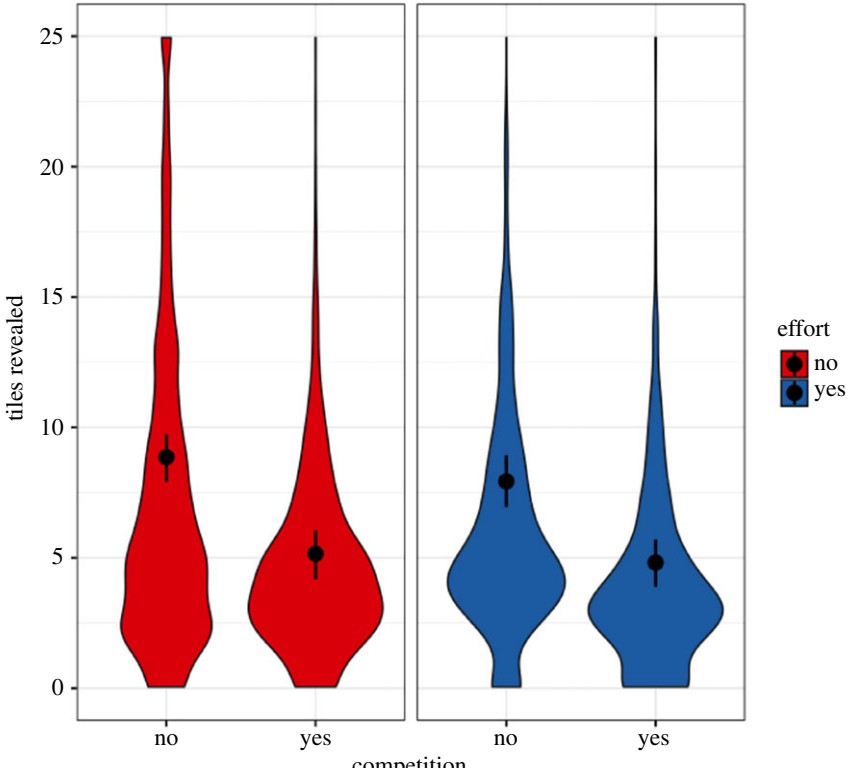

**Figure 3.** Tiles revealed. Raw data, model-predicted means and 95% HPDIs from Model 1. Participants revealed fewer tiles when in competition, in both the No-Effort and Effort conditions (95% HPDI: $(-5.03, -2.39)$, $\beta = -3.70$ and 95% HPDI: $(-4.47, -1.86)$, $\beta = -3.11$, respectively). There was no evidence for a Competition × Effort interaction on the number of tiles revealed (95% HPDI: $(-1.23, 2.54)$, $\beta = 0.60$).

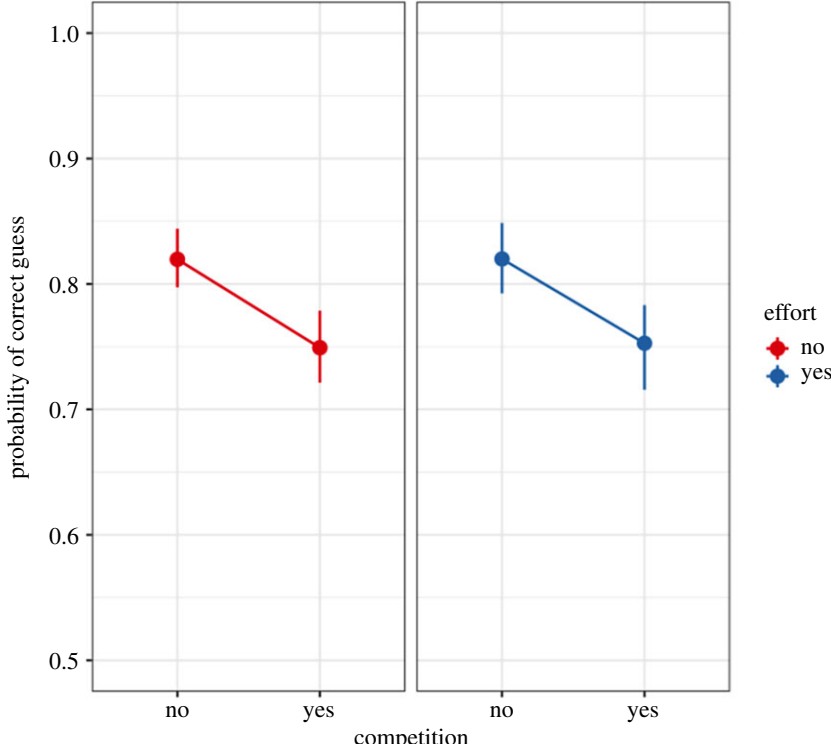

**Figure 4.** Accuracy. Raw data, model-predicted means and 95% HPDIs from Model 2. Participants were less accurate when in competition, in both the No-Effort and Effort conditions (95% HPDI: $(-0.11, -0.03)$, $\beta = -0.07$ and 95% HPDI: $(-0.11, -0.02)$, $\beta = -0.07$, respectively). There was no evidence for a Competition × Effort interaction on participant accuracy (see main text).

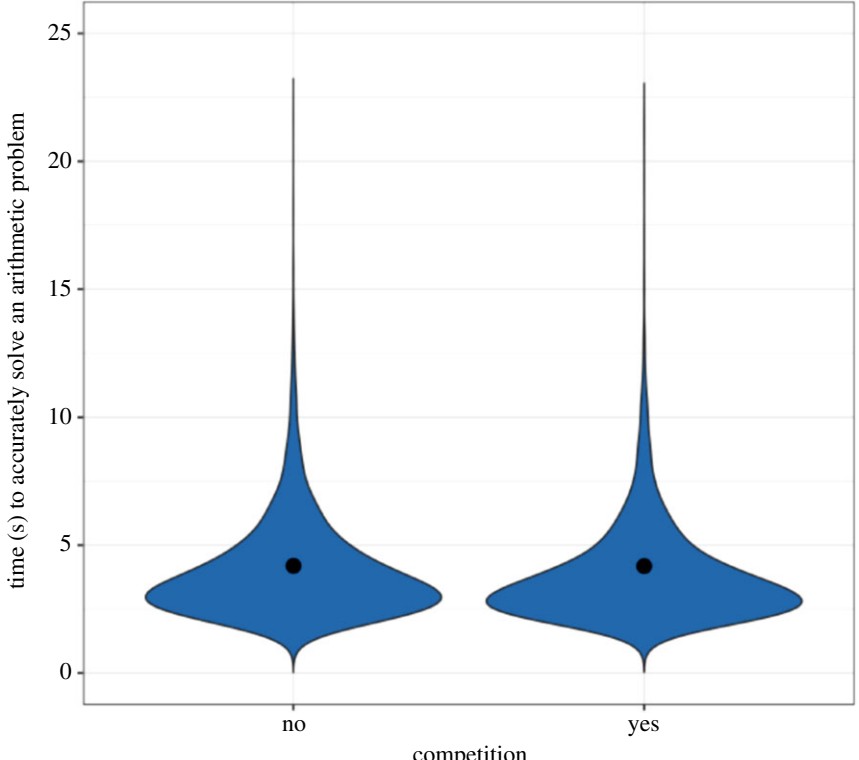

**Figure 5.** Time (s) to accurately solve one arithmetic problem. Raw data, model-predicted means and 95% HPDIs from Model 3. There was no evidence that competition increased participant effort: participants in the Competition × Effort treatment did not solve arithmetic problems more quickly than participants in the No-Competition × Effort treatment (95% HPDI: ($-0.40$, 0.35), $\beta = -0.02$).

(*H2, H3a, H3b*). To evaluate the extent to which the data provide evidence for null hypotheses of 'no effect', we calculated Bayes factors via Bayesian information criterion (BIC) approximation using frequentist statistical models with and without predictor variables relevant to the aforementioned hypotheses [41]. In almost all cases, Bayes factors provided strong to extreme support for null models without Competition as a predictor of arithmetic-problem solving times and without a Competition × Effort interaction as a predictor of tiles revealed and accuracy (electronic supplementary material). We also calculated the widely applicable information criterion (WAIC, [42]) for Bayesian implementations of statistical models with the same likelihood functions. Using information criteria to compare statistical models is one way to determine the probability that a model will best predict data from the same data-generating process, as judged by out-of-sample deviance [43]. For statistical models predicting tiles revealed and accuracy, WAIC did not provide strong evidence for or against models with a Competition × Effort interaction (electronic supplementary material, tables S3 and S4). For statistical models predicting arithmetic-problem solving time, WAIC did not provide strong evidence for or against models with Competition as a predictor (electronic supplementary material, table S5).

We explored the relationship between the effect size on a grid and the number of tiles revealed by participants, as well as their accuracy. Participants revealed fewer tiles on grids with larger effect sizes (electronic supplementary material, figure S13) and had a higher probability of correctly guessing the majority colour on grids with larger effect sizes (electronic supplementary material, figure S14). There was also evidence for several interactions (electronic supplementary material, tables S7 and S8). Model comparison using WAIC provided no evidence for a Competition × Sex interaction and Bayes factors provided strong evidence against statistical models with this interaction (electronic supplementary material). We also explored the effect of competition and effort on participants' earnings per unit time. Participants earned less money per unit time in the Effort conditions than in the No-Effort conditions, but there was no evidence that participants earned less money per unit time in the Competition treatments than in the No-Competition treatments (electronic supplementary material, figure S15a,b).

# 8. Discussion

We developed a laboratory experiment to test how competition for priority affects information sampling and accuracy in a game that mirrors aspects of scientific investigation. Following our pre-specified experimental protocol, we collected data from 260 students at Arizona State University. Quality checks indicated that the experimental manipulations were successful. Confirmatory analyses indicated that competition caused participants to make guesses with less information, thereby reducing their accuracy. Confirmatory analyses provided no evidence that competition increased participant effort (i.e. arithmetic-problem solving speed) and provided no evidence for a Competition × Effort interaction.

Our experiment intended to capture several features of scientific investigation: individuals gather data to acquire information about the world, face tradeoffs between publishing quickly and increasing accuracy by acquiring more data and compete with others for priority of publication. In this setting, participants could use two strategies to beat their opponent. One strategy is to save time by gathering less information before guessing: this increases the probability of guessing before an opponent but also increases the probability of guessing incorrectly. The other strategy is to work harder and gather information more efficiently by solving arithmetic-problems more quickly. This increases the probability of beating an opponent without compromising accuracy (e.g. a participant who reveals eight tiles in 10 s has the same expected accuracy as a participant who reveals eight tiles in 20 s). Both strategies have parallels in real-world science. The first strategy would be to invest less time in research before publishing. This might mean collecting less data and producing work with low information-value [44] or performing inadequate quality checks and increasing the probability of making a mistake [45]. The second strategy would be to gather data more efficiently than a competitor by increasing research effort. This might mean optimizing workflow, working more hours per day, hiring more research assistants or spending a smaller proportion of working time on social media [17]. Our results show that, within our simplified instantiation of the scientific process, competition leads individuals to gather less information before guessing. As expected, this strategy decreases individuals' guessing accuracy and so negatively impacts the proportion of published findings that are true. Our results thus provide evidence for the hypothesis that competition for priority can reduce research quality by encouraging individuals to trade accuracy for speed. This mirrors the findings of a similar experiment on information search in competitive environments [46]: incentivizing novel findings can create a 'race' where the individual who first produces a solution has a higher probability of winning, even if that solution is based on inadequate data. We found no evidence that competition leads individuals to acquire information faster, despite high statistical power. Thus, our experiment provides no evidence for a hypothesized positive effect of competition on individual effort.

It is worth noting that previous research has found positive effects of competition on individual effort in related paradigms [19,30,32,33,35]. We can only speculate as to why our results differ from past research. One possibility is that participants in the No-Competition × Effort treatment were already incentivized to solve arithmetic problems as quickly as possible: solving more problems allows participants to attempt more grids or guess with more information per grid, both of which increase participants' expected earnings. However, we think that this explanation is unlikely, because prior experiments have found positive effects of competition on arithmetic-problem solving rates given similar incentive structures [32,33]. An alternative explanation relates to a peculiarity of our experimental design. While participants in previous experiments typically performed a single effort task (e.g. navigating a maze, solving arithmetic problems, positioning a slider [30,33,35]), participants in our Effort condition addressed two intermingled tasks: solving arithmetic problems and guessing which tile-colour was in the majority. As a consequence, participants could adjust their behaviour in two ways in response to competition: guess earlier and/or solve arithmetic problems faster. Participants in our experiment may have favoured the former strategy because guessing earlier arguably requires less effort than solving arithmetic problems faster. If this interpretation is correct, it suggests that competition is more likely to positively affect individuals' effort when adjusting effort is the only way to outperform competitors. However, other factors could also plausibly explain the observed pattern, so the above interpretation should be taken cautiously. For instance, our arithmetic task could have been too easy (thereby generating a floor effect) to result in a notable difference in solving times between treatments. Future research could evaluate the hypothesis that competition leads individuals to preferentially adjust their behaviour so as to minimize effort by varying the extent to which individuals can invest in alternative strategies.

Our experiment did not address whether competition harms or benefits science as a whole: testing that hypothesis would require evaluating population-level outcomes (e.g. the efficiency of discovering true relationships [47]). For tractability, our instantiation of the scientific process left out many factors that exist in real-world scientific research. For instance, we only tested the direction of the effect of competition on information sampling: assuming that people compete against individuals from a non-competitive world, do people acquire more or less information on research problems? Our design thus resembled a 'game against nature' [48] and did not capture the strategic component of many competitive settings. By contrast, in real-world priority races, all parties are incentivized to adjust their behaviour based on what they know about the number of competitors and those competitors' progress [25,45,49]. Evaluating behaviour given multi-sided strategic interactions would be a useful extension of our experiment and could reveal important and complementary insights (e.g. equilibrium behaviour: the stable long-run outcome when all competitors can strategize based on expectations of opponents' likely behaviours). Furthermore, some of the factors we left out might mitigate the negative effects of competition in the real world. For example, we assumed that all research questions were independent and their solutions equally valuable. By contrast, real-world science involves interconnected questions [47,50] and solving important problems is difficult without first solving intermediate ones [22,23,25]. For example, it would have been nearly impossible for James Watson and Francis Crick to produce an accurate model of DNA's helical structure without Rosalind Franklin's X-ray diffraction images and Erwin Chargaff's experiments demonstrating equal ratios of guanine : cytosine and adenine : thymine in DNA [49]. Interdependent research questions increase the cost of mistakes because getting a single problem wrong decreases the probability of solving subsequent problems. We suspect that an experiment where the probability of solving a high-level problem depends on one's prior success at solving lower-level problems would cause people to be more careful and acquire more information, even given incentives for novel results. Our experiment also forced participants to solve well-defined problems with *a priori* correct answers. This misses the creative component of science [20]. Real-world science also involves problem choice and the ability to abandon problems without publishing a solution: scientists can choose research problems with few competitors or can 'opt out' of problems once they receive information that a competitor has made substantial progress [25]. Thus, competition in real-world science offers individuals more options than the two in our experiment (guessing earlier or increasing effort). Under more realistic conditions, individuals might be able to respond to competition in ways that do not compromise the quality of their work [51].

Despite these limitations, our experiment clearly provides proof of concept that competition for priority can reduce research quality by encouraging individuals to trade accuracy for speed. Within our simplified instantiation of the scientific process, competition caused individuals to reduce their accuracy even though inaccuracy was costly. Indeed, an incorrect guess resulted in participants losing as many points as they gained from a correct answer. By contrast, current incentives reward scientists for total number of publications, independent of their accuracy [52]. Further, the expected real-world cost of a false finding is probably lower than the benefit of publishing any novel result: in Psychology, most published studies are never replicated and studies that fail to replicate continue to accumulate citations [53]. Even given exorbitant costs for producing a finding that fails to replicate, there is a time-lag between when scientists are rewarded (e.g. upon publication) and penalized (e.g. whenever other scientists fail to replicate the study), and penalization is stochastic [14]. Although our experiment did not address these or many other aspects of science, our game provides a starting point for scholars who wish to do so in the future.

Incentive structures affect how scientists conduct research. This much is obvious. Less obvious is precisely how they do so. Currently, arguments about the effects of specific incentives on the scientific process lack a solid theoretical and empirical foundation. Our experiment takes one step towards filling the empirical gap. However, Metascience would benefit from far more investment in experimental tests of hypotheses about the effects of incentive structures on the scientific process. Filling the theoretical gap will require building formal models of the scientific process under different incentive-structure regimes. Currently, only a handful of such models exist. Without formal models, we will be forced to rely on verbal arguments. But verbal arguments lack transparency [54–56] and we may end up testing hypotheses that are logically incoherent or only make sense under a narrow range of conditions. We should strive to build a scientific discipline that is just as committed to the transparency of theory as it is to transparency of empirics. Doing so provides our best hope for improving the efficiency and reliability of science.

## 8.1. Constraints on generality

We provide a description of the constraints on generality (COG) [57] in the electronic supplementary material.

Ethics. Permission to perform this study was granted by the Arizona State University Institutional Review Board (IRB), code: STUDY00007691. All participants provided informed consent.

Data accessibility. This study was awarded Stage 1 in-principle acceptance on 31 August 2018. The Stage 1 protocol, including pilot data, can be found here: https://osf.io/24v9k/. All data, materials and code for the final Stage 2 submission can be found here: https://osf.io/7vbj9/.

Authors' contributions. L.T. and M.D. developed the study concept, design and materials. M.D. programmed the game. L.T. ran the study, conducted the analyses and drafted the manuscript. M.D. provided critical revisions. Both authors approved the final version of the manuscript.

Competing interests. We declare we have no competing interests.

Funding. Funding for this study was provided by the Arizona State University School of Human Evolution and Social Change via a research grant awarded to L.T. M.D. is supported by the European Union's Horizon 2020 research and innovation programme under Marie Sklodowska-Curie grant agreement number 748310.

Acknowledgements. We are grateful to Daniel Hruschka, Robert Boyd, Thomas Morgan, Willem Frankenhuis and Anne Scheel for extensive feedback on experimental design and previous drafts of this paper. Thanks to Thomas Morgan for his assistance with Bayesian statistical analyses, to Peder M. Isager for comments on a previous draft and to Daniel Lakens and the Red² laboratory for productive discussion of approaches to determining ROPE boundaries. We are also grateful to Christopher Chartier, Adam Sparks and an anonymous reviewer for critical feedback on previous versions of this manuscript.

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
