## [Reviewer comments · Royal Society Open Science]

Review History

RSOS-180934.R0 (Original submission)

Review form: Reviewer 1

Is the language acceptable?

Yes

Do you have any ethical concerns with this paper?

No

Have you any concerns about statistical analyses in this paper?

No

Recommendation?

Accept with minor revision

Comments to the Author(s)

This is a nice, clean experiment to assess the extent to which incentives for individual success and competition drive behavior in a scenario designed to mimic some aspects of the search for scientific knowledge.

The experimental design seems sound. I have no problems with it. I only have a few issues with the description in terms of clarity (see below).

The statistical analyses proposed seem quite good to me. In general, this design is clean without too many parameters, so the analysis should be fairly straightforward. The proposed design looks thoughtful.

The authors also present a mathematical model. The model examines only the expected payoff for the competition, no-effort condition. The motivation for this is not totally clear. I really appreciate the attempt, and I think the model is a good idea, but the execution is problematic. Most critically, the model is not fully described. The full equation(s) used to calculate each square in Figure 2 should be provided and derived. It also doesn't make sense to me that you could have a high expected payoff by revealing the exact same number of tiles as the opponent but a near-zero expected payoff if you reveal one more. Maybe I'm missing something, but the model assumptions are not clearly laid out, so I can't tell. Furthermore, the authors should be clear how the model adds to the strength of their argument(s).

OTHER COMMENTS

Line 68: "Robert Merton noted how this norm benefits science"
You might change "noted" to "argued", since it's an arguable comment.

Line 82: "there is little experimental evidence for this hypothesis."
Clarify exactly what the hypothesis is.

Lines 151-2: "Previous participants will be sampled without replacement: each participant will compete against one unique previous participant."
This is a bit confusing. Are we assuming all new participants in the competition condition? You do clarify this in the Procedure section, but some more detail here could avoid temporary confusion.

Hypotheses H1b and H3b are mathematical certainties given sufficient statistical power. They are important quality checks, but it's weird to see them phrased as hypotheses. It's like saying "If we confirm that men are taller than women on average, then we hypothesize that 10 men all stacked on each other's shoulders will be taller than 10 women stacked similarly."
You might consider rephrasing these predictions.

Review form: Reviewer 2

Is the language acceptable?

Yes

Do you have any ethical concerns with this paper?

No

Have you any concerns about statistical analyses in this paper?

No

Recommendation?

Major revision

Comments to the Author(s)

All comments in attached document (Appendix A).

Review form: Reviewer 3

Is the language acceptable?

Yes

Do you have any ethical concerns with this paper?

No

Have you any concerns about statistical analyses in this paper?

I do not feel qualified to assess the statistics

Recommendation?

Accept in principle

Comments to the Author(s)

To the authors:

Basic examination of the effects of the incentive structures of science on the quality of scientific work is extremely important; I thank the authors for their efforts.

The authors propose a very reasonable experimental approach to understanding how competition for priority influence information sampling; their method is a useful model of the games scientists play and is amenable to modification for future work to address questions raised by their eventual results of the present proposal. The specific methods to be implemented here are clearly specified and I do not see much potential for undisclosed flexibility. While I am not fully qualified to assess their statistical methods, their approach seems rigorous and well thought-out. I recommend the proposal be accepted in principle.

I do raise one substantive concern for the authors to consider. Regarding methods design whereby players in the "competition" treatment are competing against previous players from the no-competition treatment, the authors argue: "By only allowing one player to react to a past participants' performance, our design provides a conservative test of the effect of competition. In more realistic scenarios, both participants should react to each other's' behavior, resulting in larger effects of competition on individuals' decisions." This design choice is reasonable for practical reasons, but I wonder about the claim that the design is conservative because only one player can react. This might be true, but I wonder if it always must be true and whether the 1-on-1 matching method might introduce a similar type of stochasticity that you try to avoid by using the same grid sequence for each participant. I wonder if the randomly-chosen competitor – who basically has a fixed strategy – could actually have a bigger effect on the real-time player than would a two-real-time-players design. For example, if I happen to be matched against an unusually quick-and-accurate competitor, I can presumably realize this and then my competitive

disadvantage incentivizes me to make faster-, lower-information guesses as compared to when matched against a slower-less-accurate competitor. If my faster competitor was then also reacting in (near-)real time to my strategy, (s)he might even then be incentivized to slow down a bit and focus on even better accuracy, which in turn might make me slow back down a bit. Basically my first pass intuition about this type of competitive scenario suggests the possibility that a real competitive dynamic between two competitors could possibly actually result in less extreme total deviations from their non-competitive strategy – if only one of the competitors can actually adjust, isn't that one incentivized to adjust massively, rather than both of them drifting more subtly towards a middle ground? My intuition is that a different matching method -- randomly re-matching (with replacement) for every grid -- might lead players to shift more directly towards something like their optimized general competitive strategy, rather than tailoring a specific strategy to exploit a specific unchanging opponent. Again, this is all just my intuition – something for the authors to consider if they haven't yet. (The authors might be able to simulate these dynamics since expected payoffs can be calculated under various combinations of parameters.) Anyway, the proposed method seems like a reasonable starting point even if it doesn't necessarily provide the conservative test that is currently argued. Regardless of the matching strategy used, I would hope to see the implications and dynamics thoroughly discussed in the final report.

Finally, just some nitpicky concerns about language precision:

Line 84: worth specifying that the detrimental effects are detrimental to *group welfare*

Line 93: "We develop a simple experiment to test the effect of competition for priority on research output." The experiment simulates this, but it doesn't really test *research output*, does it? The scare quotes ("publish") in line 103 seem to capture this idea.

Decision letter (RSOS-180934.R0)

19-Jul-2018

Dear Dr Tiokhin,

The Editors assigned to your Stage Registered Report ("An experimental test of the effects of competition for priority on information sampling") have now received comments from reviewers. We would like you to revise your paper in accordance with the referee and editors suggestions which can be found below (not including confidential reports to the Editor). Please note this decision does not guarantee eventual acceptance.

When submitting your revised manuscript, you must respond to the comments made by the referees and upload a file "Response to Referees" in "Section 2 - File Upload". Please use this to document how you have responded to the comments, and the adjustments you have made. In order to expedite the processing of the revised manuscript, please be as specific as possible in your response.

Kind regards,
Ms Thadcha Retneswaran
Royal Society Open Science
openscience@royalsociety.org

on behalf of Chris Chambers (Registered Reports Editor, Royal Society Open Science)
openscience@royalsociety.org

Associate Editor Comments to Author (Professor Chris Chambers):

Associate Editor: 1

Comments to the Author:

Three reviewers have now assessed the manuscript. The assessments are overall positive and point to the importance and timeliness of the work, however they also raise issues that cover a wide range of the Stage 1 criteria.

To summarise the main points raised: First, as noted by Reviewer 1 please clarify the rationale (and details) of the mathematical modelling (lines 294-335). This appears to serve a generative function rather than testing the stated hypotheses and it isn't clear (yet) whether it is a necessary component to the submission. Second, Reviewer 2 raises the important issue of differences between the competitive scenario in the proposed design and real life competition in science. This may require some reframing of the proposal. Third, Reviewer 3 asks whether the competitive model is in fact more conservative than real-time competition - a question which also warrants careful consideration.

In addition, in my own reading:

- Exclusion criteria: please make crystal clear what criteria will be applied in the preregistered study (are these same as in the pilot?)
- The results of the pilot study need to be presented more coherently and made more readable, or alternatively summarised briefly but placed in supplementary information. At present they appear more as a technical readout than a manuscript.
- The power analysis should be moved from supplementary information into the main text
- line 530, line 794: should "H2c" be simply "H2"?
- line 438: suggest renaming "Confirmatory Predictions" to "Confirmatory analysis plans", as the predictions are specified earlier and this apparent duplication could otherwise be confusing. This section (importantly) maps those predictions on to the specific analyses so it is ok to restate the predictions provided the section is clearly labelled.
- Could the presentation of priors be tabulated to improve readability? I am not suggesting that any information be cut, but perhaps it could be organised more coherently, especially given broad readership of the journal.

- Formatting overload: on a purely presentational note, there is perhaps an over-abundance of heading levels (I counted: capital bold, sentence case bold, underlined regular, italics, underlined italics). Can this be simplified?

Please also be sure to address the full set of additional comments raised by all three reviewers.

Comments to Author:

Reviewer: 1

Comments to the Author(s)

This is a nice, clean experiment to assess the extent to which incentives for individual success and competition drive behavior in a scenario designed to mimic some aspects of the search for scientific knowledge.

The experimental design seems sound. I have no problems with it. I only have a few issues with the description in terms of clarity (see below).

The statistical analyses proposed seem quite good to me. In general, this design is clean without too many parameters, so the analysis should be fairly straightforward. The proposed design looks thoughtful.

The authors also present a mathematical model. The model examines only the expected payoff for the competition, no-effort condition. The motivation for this is not totally clear. I really appreciate the attempt, and I think the model is a good idea, but the execution is problematic. Most critically, the model is not fully described. The full equation(s) used to calculate each square in Figure 2 should be provided and derived. It also doesn't make sense to me that you could have a high expected payoff by revealing the exact same number of tiles as the opponent but a near-zero expected payoff if you reveal one more. Maybe I'm missing something, but the model assumptions are not clearly laid out, so I can't tell. Furthermore, the authors should be clear how the model adds to the strength of their argument(s).

OTHER COMMENTS

Line 68: "Robert Merton noted how this norm benefits science"
You might change "noted" to "argued", since it's an arguable comment.

Line 82: "there is little experimental evidence for this hypothesis."
Clarify exactly what the hypothesis is.

Lines 151-2: "Previous participants will be sampled without replacement: each participant will compete against one unique previous participant."
This is a bit confusing. Are we assuming all new participants in the competition condition? You do clarify this in the Procedure section, but some more detail here could avoid temporary confusion.

Hypotheses H1b and H3b are mathematical certainties given sufficient statistical power. They are important quality checks, but it's weird to see them phrased as hypotheses. It's like saying "If we confirm that men are taller than women on average, then we hypothesize that 10 men all stacked on each other's shoulders will be taller than 10 women stacked similarly."
You might consider rephrasing these predictions.

Reviewer: 2

Comments to the Author(s)

All comments in attached document

Reviewer: 3

Comments to the Author(s)

To the authors:

Basic examination of the effects of the incentive structures of science on the quality of scientific work is extremely important; I thank the authors for their efforts.

The authors propose a very reasonable experimental approach to understanding how competition for priority influence information sampling; their method is a useful model of the games scientists play and is amenable to modification for future work to address questions raised by their eventual results of the present proposal. The specific methods to be implemented here are clearly specified and I do not see much potential for undisclosed flexibility. While I am not fully qualified to assess their statistical methods, their approach seems rigorous and well thought-out. I recommend the proposal be accepted in principle.

I do raise one substantive concern for the authors to consider. Regarding methods design whereby players in the "competition" treatment are competing against previous players from the no-competition treatment, the authors argue: "By only allowing one player to react to a past participants' performance, our design provides a conservative test of the effect of competition. In more realistic scenarios, both participants should react to each other's behavior, resulting in larger effects of competition on individuals' decisions." This design choice is reasonable for practical reasons, but I wonder about the claim that the design is conservative because only one player can react. This might be true, but I wonder if it always must be true and whether the 1-on-1 matching method might introduce a similar type of stochasticity that you try to avoid by using the same grid sequence for each participant. I wonder if the randomly-chosen competitor – who basically has a fixed strategy – could actually have a bigger effect on the real-time player than would a two-real-time-players design. For example, if I happen to be matched against an unusually quick-and-accurate competitor, I can presumably realize this and then my competitive disadvantage incentivizes me to make faster-, lower-information guesses as compared to when matched against a slower-less-accurate competitor. If my faster competitor was then also reacting in (near-)real time to my strategy, (s)he might even then be incentivized to slow down a bit and focus on even better accuracy, which in turn might make me slow back down a bit. Basically my first pass intuition about this type of competitive scenario suggests the possibility that a real competitive dynamic between two competitors could possibly actually result in less extreme total deviations from their non-competitive strategy – if only one of the competitors can actually adjust, isn't that one incentivized to adjust massively, rather than both of them drifting more subtly towards a middle ground? My intuition is that a different matching method -- randomly re-matching (with replacement) for every grid -- might lead players to shift more directly towards something like their optimized general competitive strategy, rather than tailoring a specific strategy to exploit a specific unchanging opponent. Again, this is all just my intuition – something for the authors to consider if they haven't yet. (The authors might be able to simulate these dynamics since expected payoffs can be calculated under various combinations of parameters.) Anyway, the proposed method seems like a reasonable starting point even if it doesn't necessarily provide the conservative test that is currently argued. Regardless of the matching strategy used, I would hope to see the implications and dynamics thoroughly discussed in the final report.

Finally, just some nitpicky concerns about language precision:

Line 84: worth specifying that the detrimental effects are detrimental to *group welfare*

Line 93: "We develop a simple experiment to test the effect of competition for priority on research output." The experiment simulates this, but it doesn't really test *research output*, does it? The scare quotes ("publish") in line 103 seem to capture this idea.

Respectfully,

Author's Response to Decision Letter for (RSOS-180934.R0)

See Appendix B.

RSOS-180934.R1 (Revision)

Review form: Reviewer 1

Is the language acceptable?

Yes

Do you have any ethical concerns with this paper?

No

Have you any concerns about statistical analyses in this paper?

No

Recommendation?

Accept in principle

Comments to the Author(s)

The authors have adequately responded to my concerns. I think this is a well designed study and I look forward to seeing the results.

Review form: Reviewer 2

Is the language acceptable?

Yes

Do you have any ethical concerns with this paper?

No

Have you any concerns about statistical analyses in this paper?

No

Recommendation?

Accept in principle

Comments to the Author(s)

The authors have thoroughly responded to my initial concerns and criticisms. Instead of opening new concerns (not that I had any anyways), I have confined my comments here to authors' responses to those points I raised in my initial review.

1. I commend the authors for the inclusion of a COG statement. Great addition! I think that a COG statement is nothing to tuck away, and suggest it be included in the main text of the discussion section of the eventual manuscript.
2. I appreciate the edits to the intro and feel they do a better job of clearly communicating the potential contribution of the work.
3. Regarding my other concerns not addressed in the introduction, consider me satisfied by the point-by-point review responses provided the authors, BUT, this is based on the authors' assurances of covering considerable constraints and limitations in the discussion section. I think this is a fine plan, and I look forward to reading and providing feedback on these sections of text during stage 2 review, assuming we get to that point :).

That's it. Nice work. This ms has been pleasure to review so far!

Review form: Reviewer 3

Is the language acceptable?

Yes

Do you have any ethical concerns with this paper?

No

Have you any concerns about statistical analyses in this paper?

I do not feel qualified to assess the statistics

Recommendation?

Accept in principle

Comments to the Author(s)

I continue to think this work is important, the hypotheses are reasonable, and that the methods are well-designed for testing their hypotheses and are appropriately transparent. My own minor concerns about their previous submission have been adequately addressed, and I think the authors have adequately addressed the useful feedback of other reviewers. Thus, I again recommend acceptance in principle.

Decision letter (RSOS-180934.R1)

31-Aug-2018

Dear Dr Tiokhin

On behalf of the Editor, I am pleased to inform you that your Manuscript RSOS-180934.R1 entitled "An experimental test of the effects of competition for priority on information sampling" has been accepted in principle for publication in Royal Society Open Science. The reviewers' and editors' comments are included at the end of this email.

You may now progress to Stage 2 and complete the study as approved. Before commencing data collection we ask that you:

- 1) Update the journal office as to the anticipated completion date of your study.
- 2) Register your approved protocol on the Open Science Framework (<https://osf.io/>) or other recognised repository, either publicly or privately under embargo until submission of the Stage 2 manuscript. Please note that a time-stamped, independent registration of the protocol is mandatory under journal policy, and manuscripts that do not conform to this requirement cannot be considered at Stage 2. The protocol should be registered unchanged from its current approved state, with the time-stamp preceding implementation of the approved study design. We recommend using the dedicated interface for registering Stage 1 RRs available at <https://osf.io/rr/>

Following completion of your study, we invite you to resubmit your paper for peer review as a Stage 2 Registered Report. Please note that your manuscript can still be rejected for publication at Stage 2 if the Editors consider any of the following conditions to be met:

- The results were unable to test the authors' proposed hypotheses by failing to meet the approved outcome-neutral criteria.
- The authors altered the Introduction, rationale, or hypotheses, as approved in the Stage 1 submission.
- The authors failed to adhere closely to the registered experimental procedures. Please note that any deviations from the approved experimental procedures must be communicated to the editor immediately for approval, and prior to the completion of data collection. Failure to do so can result in revocation of in-principle acceptance and rejection at Stage 2 (see complete guidelines for further information).
- Any post-hoc (unregistered) analyses were either unjustified, insufficiently caveated, or overly dominant in shaping the authors' conclusions.
- The authors' conclusions were not justified given the data obtained.

We encourage you to read the complete guidelines for authors concerning Stage 2 submissions at <http://rsos.royalsocietypublishing.org/content/registered-reports>. Please especially note the requirements for data sharing, reporting the URL of the independently registered protocol, and that withdrawing your manuscript will result in publication of a Withdrawn Registration.

Please note that Royal Society Open Science will introduce article processing charges for all new submissions received from 1 January 2018. Registered Reports submitted and accepted after this date will ONLY be subject to a charge if they subsequently progress to and are accepted as Stage 2 Registered Reports. If your manuscript is submitted and accepted for publication after 1 January 2018 (i.e. as a full Stage 2 Registered Report), you will be asked to pay the article processing charge, unless you request a waiver and this is approved by Royal Society Publishing. You can find out more about the charges at <http://rsos.royalsocietypublishing.org/page/charges>. Should you have any queries, please contact openscience@royalsociety.org.

Once again, thank you for submitting your manuscript to Royal Society Open Science and we look forward to receiving your Stage 2 submission. If you have any questions at all, please do not hesitate to get in touch. We look forward to hearing from you shortly with the anticipated submission date for your stage two manuscript.

Kind regards,

Royal Society Open Science Editorial Office
Royal Society Open Science
openscience@royalsociety.org

on behalf of Chris Chambers (Registered Reports Editor, Royal Society Open Science)
openscience@royalsociety.org

Associate Editor Comments to Author (Professor Chris Chambers):

Associate Editor: 1

Comments to the Author:

All three reviewers responded positively to the revision and Stage 1 IPA can now be awarded.

Reviewers' comments to Author:

Reviewer: 1

Comments to the Author(s)

The authors have adequately responded to my concerns. I think this is a well designed study and I look forward to seeing the results.

Reviewer: 2

Comments to the Author(s)

The authors have thoroughly responded to my initial concerns and criticisms. Instead of opening new concerns (not that I had any anyways), I have confined my comments here to authors' responses to those points I raised in my initial review.

1. I commend the authors for the inclusion of a COG statement. Great addition! I think that a COG statement is nothing to tuck away, and suggest it be included in the main text of the discussion section of the eventual manuscript.

2. I appreciate the edits to the intro and feel they do a better job of clearly communicating the potential contribution of the work.

3. Regarding my other concerns not addressed in the introduction, consider me satisfied by the point-by-point review responses provided the authors, BUT, this is based on the authors' assurances of covering considerable constraints and limitations in the discussion section. I think this is a fine plan, and I look forward to reading and providing feedback on these sections of text during stage 2 review, assuming we get to that point :).

That's it. Nice work. This ms has been pleasure to review so far!

Reviewer: 3

Comments to the Author(s)

I continue to think this work is important, the hypotheses are reasonable, and that the methods are well-designed for testing their hypotheses and are appropriately transparent. My own minor concerns about their previous submission have been adequately addressed, and I think the authors have adequately addressed the useful feedback of other reviewers. Thus, I again recommend acceptance in principle.

Author's Response to Decision Letter for (RSOS-180934.R1)

See Appendix C.

RSOS-180934.R2 (Revision)

Review form: Reviewer 2

Is the language acceptable?

Yes

Do you have any ethical concerns with this paper?

No

Have you any concerns about statistical analyses in this paper?

No

Recommendation?

Accept with minor revision

Comments to the Author(s)

This paper was a pleasure to read (again!). I have only minor suggestions.

-The methods section should edited to past tense. It is currently still written in the future tense as if it were a stage 1 submission.

-On line 3, I recommend "irreproducible" instead of "false" (little more than a stylistic preference/suggestion).

-The sentence beginning on line 107 seem redundant with other sections and could be removed.

-On line 130 I would recommend using "additional evidence" instead of "larger samples". This keeps clear what is a true feature of the game, instead of explicitly attempting to map it onto the scientific process (which seems out of place in the methods to me).

-Same with "effect size" on line 143.

-On line 630 it would be nice to link to an osf component with the stage 1 ms for any curious readers.

-The data are able to test the authors' proposed hypotheses by passing the approved outcome-neutral criteria.

-The Introduction, rationale and stated hypotheses are the same as the approved Stage 1 submission.

-The authors adhered precisely to the registered experimental procedures.

-The authors' conclusions are justified given the data.

Review form: Reviewer 3

Is the language acceptable?

Yes

Do you have any ethical concerns with this paper?

No

Have you any concerns about statistical analyses in this paper?

I do not feel qualified to assess the statistics

Recommendation?

Accept as is

Comments to the Author(s)

These data address most aspects of the original hypotheses, and the authors provide an appropriate discussion of the limitations (e.g., of their arithmetic task as a model of effort). The final version is consistent with their Stage 1 presentation, and the methods appear to conform to those registered (with disclosure of minor deviation from Stage 1 approval). The reported exploratory analyses are justified and informative, although I am not qualified to evaluate most of the specific analytic techniques. The authors discuss their results clearly, accurately, and with appropriate nuance. All data, code, and supporting materials are openly available. Accordingly, I recommend acceptance of this version of the manuscript.

I thank the authors for this fantastic work and hope the scientific community appreciates what you have done here.

Decision letter (RSOS-180934.R2)

01-Apr-2019

Dear Dr Tiokhin:

On behalf of the Editor, I am pleased to inform you that your Stage 2 Registered Report RSOS-180934.R2 entitled "Competition for novelty reduces information sampling in a research game" has been deemed suitable for publication in Royal Society Open Science subject to minor revision in accordance with the referee suggestions. Please find the referees' comments at the end of this email.

The reviewers and Subject Editor have recommended publication, but also suggest some minor revisions to your manuscript. Therefore, I invite you to respond to the comments and revise your manuscript.

Please also ensure that all the below editorial sections are included where appropriate -- if any section is not applicable to your manuscript, please can we ask you to nevertheless include the heading, but explicitly state that the heading is inapplicable. An example of these sections is attached with this email.

- Ethics statement

- Data accessibility

[http://datadryad.org/submit?journalID=RSOS&manu=\(Document not available\)](http://datadryad.org/submit?journalID=RSOS&manu=(Document not available))

- Competing interests

- Authors' contributions

- Acknowledgements

- Funding statement

Because the schedule for publication is very tight, it is a condition of publication that you submit the revised version of your manuscript within 7 days (i.e. by the 09-Apr-2019). If you do not think you will be able to meet this date please let me know immediately.

Supplementary files will be published alongside the paper on the journal website and posted on the online figshare repository (<https://figshare.com>). The heading and legend provided for each supplementary file during the submission process will be used to create the figshare page, so

please ensure these are accurate and informative so that your files can be found in searches. Files on figshare will be made available approximately one week before the accompanying article so that the supplementary material can be attributed a unique DOI.

Please note that Royal Society Open Science will introduce article processing charges for all new submissions received from 1 January 2018. Registered Reports submitted and accepted after this date will ONLY be subject to a charge if they subsequently progress to and are accepted as Stage 2 Registered Reports. If your manuscript is submitted and accepted for publication after 1 January 2018 (i.e. as a full Stage 2 Registered Report), you will be asked to pay the article processing charge, unless you request a waiver and this is approved by Royal Society Publishing. You can find out more about the charges at <http://rsos.royalsocietypublishing.org/page/charges>. Should you have any queries, please contact openscience@royalsociety.org.

on behalf of Professor Chris Chambers (Registered Reports Editor, Royal Society Open Science)
openscience@royalsociety.org

Associate Editor Comments to Author (Professor Chris Chambers):

The Stage 2 manuscript was returned to two of the three reviewers who assessed the protocol at Stage 1. Both find that the manuscript meets the Stage 2 criteria, with only minor revisions recommended by Reviewer 2.

Please disregard the following specific recommendations from Reviewer 2, not because they are not sensible, but because they would result in deviation from the approved Stage 1 text that I do not consider to be strictly necessary.

"-On line 3, I recommend "irreproducible" instead of "false" (little more than a stylistic preference/suggestion).

-The sentence beginning on line 107 seem redundant with other sections and could be removed.

-On line 130 I would recommend using "additional evidence" instead of "larger samples". This keeps clear what is a true feature of the game, instead of explicitly attempting to map it onto the scientific process (which seems out of place in the methods to me).

-Same with "effect size" on line 143."

As noted by Reviewer 2, please amend the Data Accessibility section of the Stage 2 manuscript to include a direct URL to the accepted Stage 1 protocol on the the OSF, unchanged from the point of acceptance, and stating the date that IPA was awarded.

Once these revisions are made, full acceptance should be forthcoming without requiring further in-depth review.

Comments to Author:
Reviewer: 3

Comments to the Author(s)

These data address most aspects of the original hypotheses, and the authors provide an appropriate discussion of the limitations (e.g., of their arithmetic task as a model of effort). The final version is consistent with their Stage 1 presentation, and the methods appear to conform to those registered (with disclosure of minor deviation from Stage 1 approval). The reported exploratory analyses are justified and informative, although I am not qualified to evaluate most of the specific analytic techniques. The authors discuss their results clearly, accurately, and with appropriate nuance. All data, code, and supporting materials are openly available. Accordingly, I recommend acceptance of this version of the manuscript.

I thank the authors for this fantastic work and hope the scientific community appreciates what you have done here.

Reviewer: 2

Comments to the Author(s)

This paper was a pleasure to read (again!). I have only minor suggestions.

-The methods section should be edited to past tense. It is currently still written in the future tense as if it were a stage 1 submission.

-On line 3, I recommend "irreproducible" instead of "false" (little more than a stylistic preference/suggestion).

-The sentence beginning on line 107 seems redundant with other sections and could be removed.

-On line 130 I would recommend using "additional evidence" instead of "larger samples". This keeps clear what is a true feature of the game, instead of explicitly attempting to map it onto the scientific process (which seems out of place in the methods to me).

-Same with "effect size" on line 143.

-On line 630 it would be nice to link to an OSF component with the stage 1 ms for any curious readers.

-The data are able to test the authors' proposed hypotheses by passing the approved outcome-neutral criteria.

-The Introduction, rationale and stated hypotheses are the same as the approved Stage 1 submission.

-The authors adhered precisely to the registered experimental procedures.

-The authors' conclusions are justified given the data.

Author's Response to Decision Letter for (RSOS-180934.R2)

See Appendix D.

Decision letter (RSOS-180934.R3)

08-Apr-2019

Dear Dr Tiokhin:

It is a pleasure to accept your Stage 2 Registered Report, "Competition for novelty reduces information sampling in a research game" in its current form for publication in Royal Society Open Science.

on behalf of Professor Chris Chambers (Subject Editor)
openscience@royalsociety.org

Follow Royal Society Publishing on Twitter: [@RSocPublishing](https://twitter.com/RSocPublishing)
Follow Royal Society Publishing on Facebook:
<https://www.facebook.com/RoyalSocietyPublishing.FanPage/>
Read Royal Society Publishing's blog: <https://blogs.royalsociety.org/publishing/>

Appendix A

Review of “An experimental test of the effects of competition for priority on information sampling”

Reviewer 2

1. The significance of the research question(s)

The research question is very important. Science faces pressing concerns about the sources of low replicability of published findings. Behavioral game theoretic approaches to understanding detrimental aspects of the scientific enterprise, and possible interventions to improve replicability, could be quite promising.

My criticisms of the paper are conceptual, non “fatal”, and mostly center on the lack of correspondence between the game studied here and the scientific enterprise that it attempts to model. One of the paper’s strengths (that it attempts to elucidate the complex system of science via a simple game) is also a serious weakness of the paper, at least as presently written, for several reasons:

- The game allows for no motivation other than performance. Scientists are hopefully (!!!!) at least somewhat motivated to move human knowledge closer to truth.

- In science, credit also goes to researchers asking interesting questions. This model has all the questions asked already (blue or yellow majority) and models science as nothing but a process of collecting data that is perfectly and monotonically related to conclusion accuracy.

- The game does not include a minimum threshold of evidence required for the participant to earn a payoff. In science there typically is such a threshold in place, for example $p < .05$ (the irony of suggesting this during a Registered Reports review is not lost on me...the comment is meant to apply to most current scientific studies). If we are forced to get below a (at least nominal) type 1 error rate of 5%, the question then is how valuable it is to collect additional data beyond this threshold. This difference between the game and science seems a major one to me.

- In science one often does not know of direct competition for the same finding. We instead have the vague sense that someone may also be working on the same question. This game assumes full competition between two participants for every possible finding the researcher is investigating. Further, researchers can decrease their likelihood of being scooped by selecting their research area and the specific hypotheses they test. Many move off to less studied topics just for this reason. The game forces partners to study the same effect.

- We always hope our experimental findings generalize from our sample to the larger population of interest. Here, the likely generalizability seems particularly unlikely, because you

have such a specific and unique population of interest (scientists). Undergrads are serving as a sample to model the behavior of career scientist. Framing this study as illuminating how science operates seems quite a stretch to me. So, I'd like to see that connection "softened" substantially in the intro.

- The lack of synchrony (even pseudo synchrony) robs the game of an important element of the real scenario it attempts to model. When one is scooped in science, one finds out while still conducting one's work. The rug is pulled out in real time. This experience could be quite critical in determining the effect of scooping competition on future behavior.

- There are negative implications of being wrong (publishing a false positive) in science. You don't simply lose the points on that "round", as is the case in the game, your reputation is damaged.

-Not all "guesses" are published in science. Peers review your guesses and the evidence upon which they are based. The game assumes that all correct guesses, even those made on very little evidence, will lead to payoffs.

Given these concerns, I suggest the paper could be reframed to suggest that its findings would simply contribute to our basic understanding of information search and how priority competition affects such processes. As currently framed, the paper attempts to "take one step towards understanding how competitive incentives affect scientific reliability," and I find the link a bit too tenuous. Alternatively, for the connection to be made as explicitly as it is, I would suggest a more extensive accounting of the consistencies and inconsistencies between the game and the process of science it attempts to model.

One additional note for possible inclusion in the intro. Publishing itself developed as a way of establishing priority. There could be benefits to priority incentives because it could lead to benefits of sharing early on in the research life-cycle (pre-prints, ideas, etc.). I would prefer some discussion of these benefits.

2. The logic, rationale, and plausibility of the proposed hypotheses

The hypotheses flow logically, rationally, and plausibly from the introductory material.

3. The soundness and feasibility of the methodology and analysis pipeline (including statistical power analysis where applicable)

The basics of the proposed analysis plan appear strong. Solid exclusion criteria in place. I am concerned with the power analyses being based on data from a pilot study of only 48 participants, but plead ignorance beyond that, as the analysis model lies somewhat outside of my personal expertise.

4. Whether the clarity and degree of methodological detail would be sufficient to replicate exactly the proposed experimental procedures and analysis pipeline

Yes. A definite strength of the submission. Procedures are clearly and precisely described. Combined with the materials from the Pilot (made available by the authors on the OSF), potential replicators have what they need to attempt a replication of this study.

5. Whether the authors provide a sufficiently clear and detailed description of the methods to prevent undisclosed flexibility in the experimental procedures or analysis pipeline

Yes. This was also a definite strength of the submission.

6. Whether the authors have considered sufficient outcome-neutral conditions (e.g. positive controls) for ensuring that the results obtained are able to test the stated hypotheses

Yes. Solid quality checks are in place.

Appendix B

Editor's Remarks

Three reviewers have now assessed the manuscript. The assessments are overall positive and point to the importance and timeliness of the work, however they also raise issues that cover a wide range of the Stage 1 criteria.

To summarise the main points raised: First, as noted by Reviewer 1 please clarify the rationale (and details) of the mathematical modelling (lines 294-335). This appears to serve a generative function rather than testing the stated hypotheses and it isn't clear (yet) whether it is a necessary component to the submission.

Thank you for noting that the model was not clearly presented. We have added a description of the rationale for generating the model (See response to Reviewer 1), have clarified the details of the mathematical modeling, and have added an additional analysis that calculates player's payoff for revealing any given number of tiles when they compete against an opponent who plays a payoff-maximizing strategy (lines 302-335; 360-374; see also Supplementary Materials).

We hope that these modifications clearly highlight the value of our model. If it is still the case that you and/or the Reviewers feel that the model will confuse readers, we are willing to move the full model from the main text into the supplementary materials.

Second, Reviewer 2 raises the important issue of differences between the competitive scenario in the proposed design and real life competition in science. This may require some reframing of the proposal.

We thank Reviewer 2 for this important point. We have reframed parts of our proposal to emphasize its focus on information sampling and have added a Constraints on Generality (COG) statement to the supplementary materials (see response to Reviewer 2). We plan on extensively discussing these differences, and the generalizability of our experiment, in the discussion section of the final report.

Third, Reviewer 3 asks whether the competitive model is in fact more conservative than real-time competition - a question which also warrants careful consideration.

We appreciate the suggestion by Reviewer 3 that our experiment may substantively differ from real-time competition (see response to Reviewer 3). We have removed the claim that our design is "conservative" and plan to extensively discuss this concern in the discussion section of the final report.

In addition, in my own reading:

- Exclusion criteria: please make crystal clear what criteria will be applied in the preregistered study (are these same as in the pilot?)

We have made clear the specific exclusion criteria for the preregistered study, and have added the following sentence to the end of the "Exclusions and Outliers" section of the main study:

“These same exclusion criteria are also used in the analysis of the pilot data (see Supplementary Materials).”

- The results of the pilot study need to be presented more coherently and made more readable, or alternatively summarised briefly but placed in supplementary information. At present they appear more as a technical readout than a manuscript.

We have more concisely summarized the results of the pilot study in the manuscript and have moved the more detailed description of the results to the supplementary materials.

- The power analysis should be moved from supplementary information into the main text

Fixed.

- line 530, line 794: should "H2c" be simply "H2"?

Fixed.

- line 438: suggest renaming "Confirmatory Predictions" to "Confirmatory analysis plans", as the predictions are specified earlier and this apparent duplication could otherwise be confusing. This section (importantly) maps those predictions on to the specific analyses so it is ok to restate the predictions provided the section is clearly labelled.

Thank you for this suggestion. We have changed this section-title accordingly.

- Could the presentation of priors be tabulated to improve readability? I am not suggesting that any information be cut, but perhaps it could be organised more coherently, especially given broad readership of the journal.

Yes, thank you for this suggestion. Priors are now listed in Table 1, under the heading “Priors”. We have also tabulated the ROPEs in Table 2, under the heading “ROPEs”.

- Formatting overload: on a purely presentational note, there is perhaps an over-abundance of heading levels (I counted: capital bold, sentence case bold, underlined regular, italics, underlined italics). Can this be simplified?

Yes. We have reduced the number of heading levels to 3: capital bold, bold and italicized.

Please also be sure to address the full set of additional comments raised by all three reviewers.

Reviewer 1

This is a nice, clean experiment to assess the extent to which incentives for individual success and competition drive behavior in a scenario designed to mimic some aspects of the search for scientific knowledge.

The experimental design seems sound. I have no problems with it. I only have a few issues with the description in terms of clarity (see below).

The statistical analyses proposed seem quite good to me. In general, this design is clean without too many parameters, so the analysis should be fairly straightforward. The proposed design looks thoughtful.

We thank Reviewer 1 for the positive assessment of our manuscript.

The authors also present a mathematical model. The model examines only the expected payoff for the competition, no-effort condition. The motivation for this is not totally clear. I really appreciate the attempt, and I think the model is a good idea, but the execution is problematic. Most critically, the model is not fully described. The full equation(s) used to calculate each square in Figure 2 should be provided and derived.

We have added a more detailed description of the model, specifically regarding how the probability of a correct guess, conditional on number of tiles revealed, is calculated (lines 318-335). Because we used simulations to calculate the average amount of information for each tile revealed, there are no additional equations to present. We have added plots of information, for each effect size and number of tiles revealed, to the supplementary materials. The full code for the simulation is available in the `Information_25Tiles.R` file here: <https://osf.io/udm8g/>.

We used this model to assess the logical coherence of H1a and check whether competition for priority actually incentivizes participants to guess with smaller amounts of evidence (lines 360-368). We describe the details of our approach, and present visualizations of the results, in the supplementary materials.

We hope that this sufficiently clarifies the model execution. We are happy to provide additional detail, either in the main text or supplementary materials, if the model description is still too opaque.

It also doesn't make sense to me that you could have a high expected payoff by revealing the exact same number of tiles as the opponent but a near-zero expected payoff if you reveal one more. Maybe I'm missing something, but the model assumptions are not clearly laid out, so I can't tell.

We have added the following sentences to clarify: "This assumes that guessing at the same time or before an opponent are equivalent, and that only guessing after an opponent results in some probability of being scooped." (lines 318 – 320)

"If a player guesses after an opponent who has revealed many tiles, a player's *EP* is low: the opponent will usually correctly guess the majority color, causing the player to obtain 0 points. If an opponent reveals very few tiles, a player receives the highest *EP* by revealing a large number of tiles. This occurs because a player who guesses before this opponent has a high probability of guessing incorrectly, whereas a player who guesses after this opponent can maximize their

probability of correctly guessing the majority color by revealing as many tiles as possible.” (lines 340-346)

Furthermore, the authors should be clear how the model adds to the strength of their argument(s).

We have added a description clarifying the rationale (lines 302-306) and conclusions (lines 360-374) of the model.

OTHER COMMENTS

Line 68: “Robert Merton noted how this norm benefits science”
You might change “noted” to “argued”, since it’s an arguable comment.

Fixed. Now written as “Robert Merton noted how this norm may benefit science”

Line 82: “there is little experimental evidence for this hypothesis.”
Clarify exactly what the hypothesis is.

Fixed. Now written as “Despite these reasonable concerns, there is little empirical evidence for the hypothesis that competitive pressures to publish cause individuals to produce lower-quality research.” (Lines 84-85)

Lines 151-2: “Previous participants will be sampled without replacement: each participant will compete against one unique previous participant.”

This is a bit confusing. Are we assuming all new participants in the competition condition? You do clarify this in the Procedure section, but some more detail here could avoid temporary confusion.

We thank Reviewer 1 for noting this. To emphasize that all treatments are between subjects, we have changed the sentence below “Treatments” as follows:

“We use a 2 x 2 between-subjects design to investigate two treatments (No Competition; Competition) and two conditions (No Effort, Effort).” (Lines 152-153)

To improve clarity, we have also changed the description for the Competition treatment in the No-Effort condition as follows: “Players compete against the performance of one previous same-sex participant from the No-Competition treatment. These competitors will be sampled without replacement: each participant will compete against the performance of one unique previous participant.” (Lines 162-165)

Hypotheses H1b and H3b are mathematical certainties given sufficient statistical power. They are important quality checks, but it’s weird to see them phrased as hypotheses. It’s like saying “If we confirm that men are taller than women on average, then we hypothesize that 10 men all

stacked on each other's shoulders will be taller than 10 women stacked similarly.”
You might consider rephrasing these predictions.

We thank Reviewer 1 for noting this. We agree that it may seem strange to have two separate hypotheses, corresponding to the effect of competition on number of tiles revealed and accuracy, respectively, because participants revealing fewer tiles is the mechanism by which competition should decrease accuracy.

However, the reason we have chosen to list these as separate hypotheses is that, even with sufficient statistical power, it is possible for H1a/H3a to be confirmed, but not H1b/H3b. Consider one such case (H1a/H1b). Competition could cause participants to reveal fewer tiles when guessing, but not have reduced accuracy, if participants use a different strategy to guess the majority color when they reveal few tiles than when they reveal many tiles. For example, when revealing few tiles, participants might guess the majority color based on the tile-color that is most frequent amongst those tiles that they have revealed (e.g. if they reveal 3 tiles and 2 are blue, they guess blue 100% of the time) but when revealing many tiles, participants use probability matching to guess the majority color (e.g. if they reveal 3 tiles and 2 are blue, they guess blue 2/3 of the time and guess yellow 1/3 of the time). Because probability matching results in reduced accuracy than guessing based solely on the most frequent color, participant accuracy could not increase (or even decrease) as participants reveal more tiles.

We do not think that this scenario is likely, but it is not impossible. As such, we think that it makes sense to list the predictions about the effect of competition on accuracy as separate hypotheses, instead of re-labeling them as manipulation checks.

We remain open to re-phrasing these predictions if the reviewers and/or editor feel that this section is still unclear.

Reviewer 2

1. The significance of the research question(s)

The research question is very important. Science faces pressing concerns about the sources of low replicability of published findings. Behavioral game theoretic approaches to understanding detrimental aspects of the scientific enterprise, and possible interventions to improve replicability, could be quite promising.

My criticisms of the paper are conceptual, non “fatal”, and mostly center on the lack of correspondence between the game studied here and the scientific enterprise that it attempts to model. One of the paper's strengths (that it attempts to elucidate the complex system of science via a simple game) is also a serious weakness of the paper, at least as presently written, for several reasons:

We agree with Reviewer 2's comment that our game is an oversimplification of the scientific enterprise and does not directly correspond to the complex system of science. As we note in the introduction, many incentives affect scientists' behavior (lines 55-59). Our experiment focuses

on just one of these. We have substantially rephrased portions of the introduction to explain how our experiment contribute to important questions within metascience, despite being unrealistic in several ways. We plan to extensively discuss these limitations in the discussion section of the final report. We have also changed the abstract to note that the emphasis of our experiment on information sampling.

- The game allows for no motivation other than performance. Scientists are hopefully (!!!!) at least somewhat motivated to move human knowledge closer to truth.

We agree that there are many factors that affect scientist's behavior, including intrinsic motivation to find the truth. Our experiment is specifically designed to test one of these factors in isolation (i.e. incentives for novel results). Other factors that might amplify or attenuate the effect of competition in more realistic situations will be discussed in the discussion of the final report.

-In science, credit also goes to researchers asking interesting questions. This model has all the questions asked already (blue or yellow majority) and models science as nothing but a process of collecting data that is perfectly and monotonically related to conclusion accuracy.

Our experiment doesn't intend to capture the scientific investigation process in its entirety (this is now clearly acknowledged in the revised introduction). Instead, it intends to measure the effect of competition on information sampling in a situation that parallels scientific investigation. This allows us to test one potential causal factor for low-quality research: competition for novel findings. Given the widespread concerns that current incentives may contribute to published findings that do not replicate, we see this as an essential first step. We hope that our modified introduction makes this point clearer (Lines 93-102).

- The game does not include a minimum threshold of evidence required for the participant to earn a payoff. In science there typically is such a threshold in place, for example $p < .05$ (the irony of suggesting this during a Registered Reports review is not lost on me...the comment is meant to apply to most current scientific studies). If we are forced to get below a (at least nominal) type 1 error rate of 5%, the question then is how valuable it is to collect additional data beyond this threshold. This difference between the game and science seems a major one to me.

We thank Reviewer 2 for raising this point. Our experiment intends to mimic a situation in which collecting additional information reduces uncertainty and brings individuals closer to the truth. The relationship between the amount of data and the robustness of the inferences made from these data is true irrespective of the $p < .05$ threshold. Adding an evidence threshold is certainly an interesting possible extension of our design, and we will consider discussing this possibility in the discussion section of the final report.

- In science one often does not know of direct competition for the same finding. We instead have the vague sense that someone may also be working on the same question. This game assumes full competition between two participants for every possible finding the researcher is investigating.

We thank Reviewer 2 for noting this important point. We plan on discussing this in the discussion section of our final report.

Further, researchers can decrease their likelihood of being scooped by selecting their research area and the specific hypotheses they test. Many move off to less studied topics just for this reason. The game forces partners to study the same effect.

Our experiment is not designed to test the effect of competition on choice of research question, although we agree that the effect of competition on choice of research question is an important topic (and one that has been the focus of recent formal models of science: <https://arxiv.org/pdf/1605.05822v2.pdf>). We plan to discuss this point in the discussion section of our final report. However, researchers do often compete for priority on the same research question (and specific concern that this may lead to rushed, low-quality research was explicitly noted as a reason for the change in editorial policy at PLOS Biology). Our experiment is designed to test whether these concerns are justified.

-We always hope our experimental findings generalize from our sample to the larger population of interest. Here, the likely generalizability seems particularly unlikely, because you have such a specific and unique population of interest (scientists). Undergrads are serving as a sample to model the behavior of career scientist. Framing this study as illuminating how science operates seems quite a stretch to me. So, I'd like to see that connection "softened" substantially in the intro.

We agree that it is important to note potential constraints on generalizability of our experiment. We have modified our introduction (see especially lines 93-102), have added a Constraints on Generality (COG) statement to the supplementary materials of our manuscript, and will discuss potential barriers to generalizability in the discussion section of our final report.

Simons, D. J., Shoda, Y., & Lindsay, D. S. (2017). Constraints on generality (COG): A proposed addition to all empirical papers. *Perspectives on Psychological Science*, 12(6), 1123-1128.

- The lack of synchrony (even pseudo synchrony) robs the game of an important element of the real scenario it attempts to model. When one is scooped in science, one finds out while still conducting one's work. The rug is pulled out in real time. This experience could be quite critical in determining the effect of scooping competition on future behavior.

We do agree that scientists sometimes experience having "the rug pulled out in real time". We initially considered designing our experiment in this way, such that players were immediately notified when another player submitted their solution. However, this design introduces various problems, as we note in the "Treatments" section of our manuscript (Lines 172-183).

In absence of a perfect way to implement competition, we consider our method as a useful starting point to test our hypotheses. We plan to extensively discuss the potential limitations of our design in the final report.

- There are negative implications of being wrong (publishing a false positive) in science. You don't simply lose the points on that "round", as is the case in the game, your reputation is damaged.

We agree that there may also be longstanding repercussions to publishing false positives (e.g. losing future collaborators, lower probability of having findings accepted for publication in the future). These will make for interesting extensions of our current design, and we will consider discussing them in the discussion section of our final report.

-Not all "guesses" are published in science. Peers review your guesses and the evidence upon which they are based. The game assumes that all correct guesses, even those made on very little evidence, will lead to payoffs.

We thank Reviewer 2 for this interesting point. We agree that this is a difference between our design and the reality of scientific publishing - incorporating peer review of research findings will be an interesting direction for future research. However, the reality of the current publishing system is that positive research findings are often published, even when they are based on very little evidence (e.g. small sample sizes). This is why retrospective power analyses across the social and behavior sciences find that published research is dramatically underpowered:

Ioannidis, J. P., Stanley, T. D., & Doucouliagos, H. (2017). The power of bias in economics research. *The Economic Journal*, 127(605), F236-F265.

Button, K. S., Ioannidis, J. P., Mokrysz, C., Nosek, B. A., Flint, J., Robinson, E. S., & Munafò, M. R. (2013). Power failure: why small sample size undermines the reliability of neuroscience. *Nature Reviews Neuroscience*, 14(5), 365.

Smaldino, P. E., & McElreath, R. (2016). The natural selection of bad science. *Royal Society Open Science*, 3(9), 160384.

Given these concerns, I suggest the paper could be reframed to suggest that its findings would simply contribute to our basic understanding of information search and how priority competition effects such processes. As currently framed, the paper attempts to "take one step towards understanding how competitive incentives affect scientific reliability," and I find the link a bit too tenuous. Alternatively, for the connection to be made as explicitly as it is, I would suggest a more extensive accounting of the consistencies and inconsistencies between the game and the process of science it attempts to model.

We hope that the modifications we made in the introduction makes it clearer that our experiment does not intend to replicate the scientific process in its entirety. Rather it intends to study how competition affects information-sampling strategies in a situation that partly mimics scientific investigation.

One additional note for possible inclusion in the intro. Publishing itself developed as a way of establishing priority. There could be benefits to priority incentives because it could lead to benefits of sharing early on in the research life-cycle (pre-prints, ideas, etc.). I would prefer

some discussion of these benefits.

We have added the following description of models of the scientific process that have addressed this question:

Lines 70 – 72: “Models of academic priority races substantiate Merton’s intuition: under some conditions, rewarding priority of discovery can incentivize scientists to disclose partial results (22–25) and can lead to efficient distributions of scientists across research problems (26).”

2. The logic, rationale, and plausibility of the proposed hypotheses
The hypotheses flow logically, rationally, and plausibly from the introductory material.

3. The soundness and feasibility of the methodology and analysis pipeline (including statistical power analysis where applicable)

The basics of the proposed analysis plan appear strong. Solid exclusion criteria in place. I am concerned with the power analyses being based on data from a pilot study of only 48 participants, but plead ignorance beyond that, as the analysis model lies somewhat outside of my personal expertise.

Within a Frequentist framework, conducting a power analysis based on effect-size estimates from a pilot study can be problematic, because it can lead to both overpowered and underpowered studies (Albers & Lakens, 2018). However, this is not an issue for power analyses within a Bayesian framework: parameter values are sampled from a probability distribution that reflects our uncertainty about the true population parameter. Large uncertainty in an effect (e.g. an uninformative pilot study) would simply result in large uncertainty about the true population parameters. For our experiment, we sample parameters from a probability distribution generated by pilot data, instead of relying on effects from previous literature, because our experiment uses a novel protocol. Doing so is standard procedure for Bayesian power analyses (Kruschke, 2014, ch. 13). In our specific case, the pilot study was sufficiently informative: we know enough about the probable underlying effect sizes that 260 participants provides 95% statistical power to detect almost all of the hypothesized effects.

Albers, C., & Lakens, D. (2018). When power analyses based on pilot data are biased: Inaccurate effect size estimators and follow-up bias. *Journal of Experimental Social Psychology*, 74, 187-195.

Kruschke, J. (2014). *Doing Bayesian data analysis: A tutorial with R, JAGS, and Stan*. Academic Press.

4. Whether the clarity and degree of methodological detail would be sufficient to replicate exactly the proposed experimental procedures and analysis pipeline

Yes. A definite strength of the submission. Procedures are clearly and precisely described. Combined with the materials from the Pilot (made available by the authors on the OSF), potential replicators have what they need to attempt a replication of this study.

5. Whether the authors provide a sufficiently clear and detailed description of the methods to prevent undisclosed flexibility in the experimental procedures or analysis pipeline

Yes. This was also a definite strength of the submission.

6. Whether the authors have considered sufficient outcome-neutral conditions (e.g. positive controls) for ensuring that the results obtained are able to test the stated hypotheses

Yes. Solid quality checks are in place.

We thank Reviewer 2 for these positive assessments of our manuscript, and for their detailed, constructive feedback on the experimental framing and design.

Reviewer 3

Basic examination of the effects of the incentive structures of science on the quality of scientific work is extremely important; I thank the authors for their efforts.

The authors propose a very reasonable experimental approach to understanding how competition for priority influence information sampling; their method is a useful model of the games scientists play and is amenable to modification for future work to address questions raised by their eventual results of the present proposal. The specific methods to be implemented here are clearly specified and I do not see much potential for undisclosed flexibility. While I am not fully qualified to assess their statistical methods, their approach seems rigorous and well thought-out. I recommend the proposal be accepted in principle.

We thank Reviewer 3 for his positive assessment of our manuscript.

I do raise one substantive concern for the authors to consider. Regarding methods design whereby players in the “competition” treatment are competing against previous players from the no-competition treatment, the authors argue: “By only allowing one player to react to a past participants’ performance, our design provides a conservative test of the effect of competition. In more realistic scenarios, both participants should react to each other’s’ behavior, resulting in larger effects of competition on individuals’ decisions.” This design choice is reasonable for practical reasons, but I wonder about the claim that the design is conservative because only one player can react. This might be true, but I wonder if it always must be true and whether the 1-on-1 matching method might introduce a similar type of stochasticity that you try to avoid by using the same grid sequence for each participant. I wonder if the randomly-chosen competitor—who basically has a fixed strategy—could actually have a bigger effect on the real-time player than would a two-real-time-players design. For example, if I happen to be matched against an unusually quick-and-accurate competitor, I can presumably realize this and then my competitive disadvantage incentivizes me to make faster-, lower-information guesses as compared to when matched against a slower-less-accurate competitor. If my faster competitor was then also

reacting in (near-)real time to my strategy, (s)he might even then be incentivized to slow down a bit and focus on even better accuracy, which in turn might make me slow back down a bit. Basically my first pass intuition about this type of competitive scenario suggests the possibility that a real competitive dynamic between two competitors could possibly actually result in less extreme total deviations from their non-competitive strategy – if only one of the competitors can actually adjust, isn't that one incentivized to adjust massively, rather than both of them drifting more subtly towards a middle ground? My intuition is that a different matching method -- randomly re-matching (with replacement) for every grid -- might lead players to shift more directly towards something like their optimized general competitive strategy, rather than tailoring a specific strategy to exploit a specific unchanging opponent. Again, this is all just my intuition – something for the authors to consider if they haven't yet. (The authors might be able to simulate these dynamics since expected payoffs can be calculated under various combinations of parameters.) Anyway, the proposed method seems like a reasonable starting point even if it doesn't necessarily provide the conservative test that is currently argued. Regardless of the matching strategy used, I would hope to see the implications and dynamics thoroughly discussed in the final report.

We thank Reviewer 3 for raising this interesting point. We agree that different matching methods might affect the results and so removed the claim that our design is “conservative” from the manuscript.

We initially intended to implement real-time interactions between competitors but quickly realized that it would generate biases (lines 173-184). The “re-matching” design is an interesting suggestion that we will consider for the future. One concern that we have with this design is that it would require many non-competitive participants to complete the study before running the competitive treatment. Running one of the two treatments earlier than the others might result in unforeseen biases (for example if the second treatment is closer to participants' exams). In absence of a perfect solution, we consider our method as a useful starting point to test our main hypothesis and plan to extensively discuss the potential limitations of our design in the final report.

Finally, just some nitpicky concerns about language precision:

Line 84: worth specifying that the detrimental effects are detrimental to *group welfare*

We thank R3 for this suggestion. The end of the introduction has been rewritten to address R2's concerns. The revised version did not allow us to implement to this specific change, but we plan to discuss the effects of competition on individual vs group welfare in the discussion of the final report.

Line 93: “We develop a simple experiment to test the effect of competition for priority on research output.” The experiment simulates this, but it doesn't really test *research output*, does it? The scare quotes (“publish”) in line 103 seem to capture this idea.

We have changed this sentence as follows: “We develop a simple experiment to test the effect of competition for priority on information-acquisition strategies.”

Appendix C

Dear editor,

Please find attached the Stage 2 Registered Report submission for Manuscript RSOS-180934.R2.

We have conducted this experiment in accordance with the plan pre-specified in the Stage 1 Registered Report submission. We have noted any deviations from this plan in the manuscript. For completeness and readability, we have made several minor modifications to the Stage 1 manuscript since its approval. These modifications are as follows:

- 1) Added the following reference in the introduction (lines 50-51): Higginson, A. D., & Munafò, M. R. (2016). Current incentives for scientists lead to underpowered studies with erroneous conclusions. *PLoS Biology*, 14(11), e2000995.
- 2) Added several references to other experiments that have tested the effect of competition on effort (lines 88-89)
 - a. Dohmen T, Falk A. Performance pay and multidimensional sorting: Productivity, preferences, and gender. *Am Econ Rev*. 2011;101(2):556–90.
 - b. Niederle M, Vesterlund L. Do women shy away from competition? Do men compete too much? *Q J Econ*. 2007;122(3):1067–1101
- 3) Slightly modified lines 122-124 to more clearly introduce the following sections of the manuscript.
- 4) Moved the theoretical model section above the section that introduces hypotheses. This is a clearer structure, because Hypothesis 1 follows from the theoretical model. We have also added “See Hypotheses Below” on line 271 to provide a link between the theoretical model and the hypotheses section.
- 5) We have added three footnotes (pg 10, pg 16, pg 19). The first and third indicate a pre-approved modification to the Stage 1 plan. The second notes a peculiar assumption of the theoretical model, and point the reader to a different theoretical analysis in the SI.
- 6) We have changed the word “player” to “participant” throughout the manuscript, for consistency.
- 7) We have added the following text to Table 1: “Gamma distributions are defined by parameters for shape and rate. Normal distributions are defined by parameters for mean and standard deviation”. This clarifies the meaning of the parameters in the table.

We look forward to your evaluation.

Sincerely,

Leonid Tiokhin and Maxime Derex

Appendix D

Dear editor,

Thank you for your email. We are happy to hear that our Stage 2 submission of manuscript RSOS-180934.R2 has been deemed suitable for publication in *Royal Society Open Science*.

We have incorporated all suggested changes in our revised manuscript and look forward to your evaluation.

Sincerely,

Leonid Tiokhin and Maxime Derex